# MARIO: Model Agnostic Recipe for Improving OOD Generalization of Graph Contrastive Learning

Submission Id: 77

## ABSTRACT

In contemporary research, large-scale graphs and graph neural networks (GNNs) serve as prevalent tools for organizing and modeling web-related data. Nevertheless, the dynamic nature of web content, characterized by continual change and evolution over time (e.g., the prevailing trends and citation patterns in online citation networks), presents a formidable challenge to the adaptability of GNNs in addressing these distributional shifts. In this work, we investigate the problem of out-of-distribution (OOD) generalization for unsupervised learning methods on graph data. To improve the robustness against such distributional shifts, we propose a Model-Agnostic Recipe for Improving OOD generalizability of unsupervised graph contrastive learning methods, which we refer to as MARIO. MARIO introduces two principles aimed at developing distributional-shift-robust graph contrastive methods to overcome the limitations of existing frameworks: (i) Invariance principle that incorporates adversarial graph augmentation to obtain invariant representations and (ii) Information Bottleneck (IB) principle for achieving generalizable representations through refining representation contrasting. To the best of our knowledge, this is the first work that investigates the OOD generalization problem of graph contrastive learning, with a specific focus on node-level tasks. Through extensive experiments, we demonstrate that our method achieves state-of-the-art performance on the OOD test set, while maintaining comparable performance on the in-distribution test set when compared to existing approaches.

## CCS CONCEPTS

• **Theory of computation → Unsupervised learning and clustering**; • **Information systems → Data mining**.

## KEYWORDS

Graph Neural Networks, Domain Generalization, Self-Supervised Learning, Graph Representation Learning, Pre-Training

**ACM Reference Format:**
Anonymous Author(s). 2024. MARIO: Model Agnostic Recipe for Improving OOD Generalization of Graph Contrastive Learning. In *Proceedings of the ACM Web Conference 2024 (WWW '24)*. ACM, New York, NY, USA, 19 pages. https://doi.org/XXXXXXX.XXXXXXX

## 1 INTRODUCTION

Graph-structured data is prevalent in Web applications, including community detection [44], paper classification [5], and social recommendationn [15]. To address these tasks, GNNs have proven to be effective tools. Nevertheless, many existing graph methods assume that training and testing data follow the same distribution, which is not always the case in real-world scenarios. For example, citation networks exhibit distribution shifts due to evolving topics and citation patterns [19, 26, 64]. Beyond addressing distribution shifts, effectively utilizing massive unlabeled graph data remains a challenging problem. Thus, this paper aims to uncover principles for achieving superior out-of-distribution (OOD) generalization performance with unlabeled graph data. To this end, two primary challenges need to be addressed:

*Challenge 1:* Non-Euclidean data structure of graphs causes complex distributional shifts (feature-level and topology-level) and lack of environment labels (due to the inherent abstraction of graph), which in consequence severely qualifies the direct application of existing OOD generalization methods.

*Challenge 2:* Most existing OOD generalization methods heavily rely on label information. It remains a practical challenge how to elicit invariant representations when no access to labels is provided.

Many efforts have been made towards the resolution of the challenges above. To address *Challenge 1*, EERM [64], GIL [38], and DIR [66] employ environment generators to simulate diverse distributional shifts in graph data. By minimizing the mean and variance of risks across multiple graphs and environments, these methods manage to capture invariant features that generalize well on unseen domains. However, these approaches heavily rely on the label information, which cannot be deployed in unsupervised settings, as *Challenge 2* suggests. Regarding the second challenge, graph contrastive learning (GCL) has recently emerged as a prominent unsupervised graph learning framework. Although some of the GCL methods have demonstrated superior performance under in-distribution tests, their efficacy under out-of-distribution tests is still unclear, as they do not explicitly target on improving the OOD generalization ability. In summary, current methods struggle to effectively address both challenges simultaneously in the field of unsupervised OOD generalization for graph data.

In this work, we for the first time systematically study the robustness of current unsupervised graph learning methods [22, 25, 57, 62, 70, 73, 77, 78] while facing distribution shifts. By analyzing the common drawbacks of GCL methods, we propose a **M**odel-**A**gnostic **R**ecipe for **I**mproving **O**OD generalization of GCL methods (MARIO[1]). To solve the above challenges, MARIO works on the two crucial components of a typical GCL method, *i.e.*, view generation and representation contrasting, as depicted in Figure 1, and

---
[1]Based on this recipe, we provide a shift-robust graph contrastive framework coined as MARIO

leaves the encoding models as an open choice for existing and future works for the sake of universal application. Concretely, MARIO introduces two principles aimed at developing distributional-shift-robust graph contrastive methods to overcome the limitations of existing frameworks: (i) Invariance principle that incorporates adversarial graph augmentation to acquire invariant representations (solving *Challenge 1*) and (ii) Information Bottleneck (IB) principle for achieving generalizable representations (solving *Challenge 2*). Furthermore, IB constraint in conjunction with invariance principle can address key failures linked to invariant features [1, 36]. Throughout extensive experiments, we observe that some graph contrastive methods are more robust to distribution shift, especially in datasets with artificial spurious features (*e.g.*, GOOD-CBAS). Furthermore, our proposed model-agnostic recipe MARIO reaches comparable performance on the in-distribution test domain but shows superior performance on out-of-distribution test domain, regardless of what model is deployed for view encoding.

As the first work that investigates the efficacy of unsupervised graph learning methods while facing distribution shifts [2], our paper's main contributions are summarized as follows:

- Through extensive experiments, we observe that some GCL methods are more robust to OOD tests than their supervised counterparts, providing insights for solving the challenge of graph OOD generalization.
- Motivated by invariant learning and information bottleneck, we analyze the limitations of the main components in current GCL frameworks for OOD generalization, and we further propose a **M**odel-**A**gnostic **R**ecipe for **I**mproving **O**OD generalization of GCL methods (MARIO).
- The proposed model-agnostic recipe MARIO can be seamlessly deployed for various graph encoding models, achieving SOTA performance under the out-of-distribution test set while reaching comparable performance under the in-distribution test set.

## 2 BACKGROUND AND PROBLEM FORMULATION

In this section, we will start with the notations we use throughout the rest of the paper (Sec. 2.1); then we introduce the problem definition and background of graph OOD generalization (Sec. 2.2) and graph self-supervised learning methods (Sec. 2.3); finally, we formalize the problem of graph contrastive learning for OOD generalization (GCL-OOD) in Sec. 2.4.

### 2.1 Notations

Let $\mathcal{G}, \mathcal{Y}$ represent input and label space respectively. $f_\phi(\cdot) = p_\omega \circ g_\theta(\cdot)$ represents graph predictor which consists of a GNN encoder $g_\theta(\cdot)$ and a classifier $p_\omega(\cdot)$. The graph predictor $f_\phi : \mathcal{G} \rightarrow \mathcal{Y}$ maps instance $G = (A, X) \in \mathcal{G}$ to label $Y \in \mathcal{Y}$ where $A \in \mathbb{R}^{N \times N}$ is the adjacent matrix and $X \in \mathbb{R}^{N \times D}$ is the node attribute matrix. Here, $N, D$ denote the number of nodes and attributes, respectively. To measure the discrepancy between the prediction and the ground-truth label, a loss function $\ell_{\text{sup}}$ is used (*e.g.*, cross-entropy

___

[2]In this work, we focus on node-level downstream tasks, which are more challenging than graph-level tasks due to the interconnected nature of instances within a graph.

loss). For unsupervised learning, a pretext loss $\ell_{\text{unsup}}$ is applied (*e.g.*, InfoNCE loss [47]). And we use $\mathcal{T}$ as augmentation pool, the augmentation function $\tau$ is randomly selected from $\mathcal{T}$ according to some distribution $\pi$. Let $\mathcal{G}^{\text{tar}}$ denote downstream dataset.

### 2.2 Graph OOD Generalization

**Problem definition.** Given training set $\mathcal{G}^{\text{train}} = (G_i, Y_i)_{i=1}^N$ that contains $N$ instances drawn from the training distribution $P_{\text{train}}(G, Y)$. In the supervised setting, it aims to learn an optimal graph predictor $f^*$ that can exhibit the best generalization performance on the data sampled from the test distribution:

$$f_\phi^* = \arg\min_{f_\phi} \mathbb{E}_{G, Y \sim P_{\text{test}}} \left[ \ell_{\text{sup}} \left( f_\phi(G), Y \right) \right], \tag{1}$$

where $P_{\text{test}}(G, Y) \neq P_{\text{train}}(G, Y)$ means there exists a distribution shift between training and testing sets, the optimal predictor trained on the training set may not generalize well on the testing set.

**Related works.** Out-of-distribution generalization algorithms [37, 55] have gained prominence for handling unknown distribution shifts in response to the growing need for managing unseen data in real-world scenarios. Techniques like robust optimization [27, 53], invariant representation/predictor learning [4, 75], and causal approaches [24, 51] have been proposed to tackle these issues. In this subsection, we emphasize *invariant representation learning* for graphs due to its practical assumptions and theoretical foundation.

Graph invariant learning methods extend invariant learning on graph domain which are widely investigated recently [38, 43, 64, 66]. EERM [64], GIL[38], DIR [66] rely on environment generators to find invariant predictive patterns with labels. GSAT [43] leverages the attention mechanism and the information bottleneck principle [3] to construct interpretable GNNs for learning invariant subgraphs under distribution shifts. CIGA [11] proposes an information-theoretic objective to extract invariant subgraphs, ensuring immunity to distribution shifts. However, most works focus on graph-level tasks under supervised setting.

Dealing with node-level tasks without labels is more challenging due to the interconnected samples, large scale of graphs and lack of supervision. In this work, we aim to pioneer an OOD algorithm for these tasks, addressing this challenging problem.

### 2.3 Graph Contrastive Learning

**Problem definition.** Graph contrastive learning (GCL) is a representative self-supervised graph learning method [22, 62, 70, 77, 78]. It consists of three main components: view generation, view encoding, and representation contrasting (Figure 1). Given an input graph $G$, two graph augmentations $\tau_\alpha$ and $\tau_\beta$ are used to generate two augmented views $G_\alpha = \tau_\alpha(G)$ and $G_\beta = \tau_\beta(G)$, respectively. A GNN model $g_\theta$ [33, 61, 67] is then applied to the augmented views to produce node representations $g_\theta(G_\alpha) \in \mathbb{R}^{N \times D}$. Lastly, a contrastive loss function is applied to representations, pulling together the positive pairs while pushing apart negative pairs. Taking the InfoNCE loss [47] as an example, the formulation follows:

$$\mathcal{L}_{\text{MI}}(g_\theta; \mathcal{G}, \pi) = -\mathop{\mathbb{E}}_{G \in \mathcal{G}} \mathbb{E}_{\tau_\alpha, \tau_\beta \sim \pi^2} \left\| g_\theta(\tau_\alpha(G)) - g_\theta\left(\tau_\beta(G)\right) \right\|^2$$

$$+ \mathop{\mathbb{E}}_{G \in \mathcal{G}} \log \mathop{\mathbb{E}}_{G' \in \mathcal{G}} \mathbb{E}_{\tau' \sim \pi} \left[ e^{\|g_\theta(\tau_\alpha(G)) - g_\theta(\tau'(G'))\|^2} \right], \tag{2}$$

where $G'$ denotes a randomly sampled graph from the graph data distribution $\mathcal{G}$, serving as the constraint for non-collapsing representations. For simplicity, the representation produced by encoder is automatically normalized to a unit sphere, *i.e.*, $\|g_\theta(G)\| = 1, \forall G \in \mathcal{G}$. By minimizing this loss, the former term (aka alignment loss $\mathcal{L}_{\text{align}}$ [63]) pulls positive pairs together by encouraging their similarity, and the latter term (aka uniformity loss $\mathcal{L}_{\text{uniform}}$ [63]) pushes negative pairs apart. The quality of the pre-trained graph encoder is then evaluated by the linear separability of the final representations. Namely, a linear classifier $p_\omega$ is built on top of the frozen encoder:

$$p_\omega^* = \arg\min_{p_\omega} \mathbb{E}_{G,Y \sim P_{\text{train}}} \left[ \ell_{\sup} \left( p_\omega \circ g_\theta^*(G), Y \right) \right], \quad (3)$$

where $g_\theta^*(\cdot)$ is obtained by minimizing Equation 2 without labels. For evaluating pre-trained model, the optimal graph predictor $f_\phi^* = p_\omega^* \circ g_\theta^*$ will be applied to testing data.

**Related work.** Unsupervised contrastive methods in graph domains have recently shown impressive progress, even surpassing supervised methods in some cases [25, 62, 70, 76–78]. These self-supervised methods typically assume that training and test data share the same distribution. However, their effectiveness under real-world scenarios with distribution shifts between training and test sets remains uncertain. In response, RGCL [40] introduces a rationale generator for discovering causal subgraphs, enhancing OOD generalization within contrastive learning. Nevertheless, RGCL is not suitable for node-level tasks due to memory constraints, and finding rational subgraphs for individual nodes proves impractical. Additionally, RGCL solely focuses on enhancing view generation.

In this work, we are the first to investigate the robustness of graph self-supervised methods in the face of distribution shifts on node-level tasks. We provide a model-agnostic approach to enhance OOD generalization of GCL. In the next subsection, we will formally define the problem of graph contrastive learning for OOD generalization (GCL-OOD) and highlight its challenges.

## 2.4 GCL-OOD: Graph Contrastive Learning for OOD Generalization

Suppose $\Phi(G)$ is invariant rationales of input instance $G$ which is stable in different environments (augmentations) following invariance assumption [37, 55]:

$$\mathbb{E}\left[Y \mid \Phi\left(G_e\right)\right] = \mathbb{E}\left[Y \mid \Phi\left(G_{e'}\right)\right], \quad \forall e, e' \in \text{supp}\left(\mathcal{E}_{tr}\right), \quad (4)$$

where $\mathcal{E}_{tr}$ denotes the set of training environments[3] and the above equation represents that invariant rationales exhibit predictive invariant (stable) correlations with semantic labels across different environments.

The optimal (invariant) graph encoder $g_\theta^\star$ achieves the invariant rationales $\Phi(G)$ across all the environments [4]:

$$g_\theta^\star\left(G_e\right) = g_\theta^\star\left(G_{e'}\right) = \Phi(G), \quad \forall e, e' \in \text{supp}\left(\mathcal{E}_{tr}\right). \quad (5)$$

However, during the pre-training of GCL, we have no access to labels under self-supervised setting. Here, we build a connection between pre-text loss $\mathcal{L}_{\text{MI}}\left(g_\theta; \mathcal{G}, \pi\right)$ and downstream loss

---

[3]Data in different environments has different data distributions.
[4]We assume the augmentation function will not change the semantic labels of the original input here.

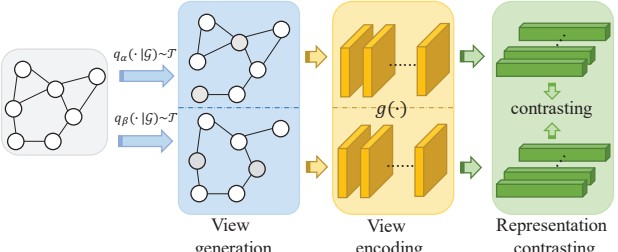

**Figure 1: The pipeline of graph contrastive learning.**

$\mathcal{R}\left(g_\theta; \mathcal{G}^{\text{tar}}\right)$ by upper-bounding referring to [28, 74]:

$$\mathcal{R}\left(p_\omega \circ g_\theta; \mathcal{G}_\pi\right) \leq c\|p_\omega\|\sqrt{K}\sigma \left(\mathcal{L}_{\text{align}}\left(g_\theta; \mathcal{G}, \pi\right)\right)^{\frac{1}{4}} + \|p_\omega\|\zeta(\sigma, \delta)$$
$$+ \sum_{k=1}^{K} \mathcal{G}_\pi\left(C_k\right)\|e_k - p_\omega \circ \mu_k\left(g_\theta; \mathcal{G}_\pi\right)\|, \quad (6)$$

where $c$ is a positive constant, $\zeta(\sigma, \delta)$ is a set of constants that only depends on $(\sigma, \delta)$-augmentation [28], $C_k \subseteq \mathcal{G}$ is the set of the data points in class $k$, $\mu_k(g_\theta; \mathcal{G}) := \mathbb{E}_{G \sim \mathcal{G}}[g_\theta(G)]$ for $k \in [K]$. The derivation and more illustrations can be found in Appendix A.

The first term in Equation 6 is the alignment loss optimized during pre-training on $\mathcal{G}$. The second term is determined by the $(\sigma, \delta)$ quantity of the data augmentation, with larger $\sigma$ and smaller $\delta$ resulting in smaller $\zeta(\sigma, \delta)$. The third term is associated with the linear layer $p$ and is minimized in downstream training. The class centers can be distinguished by choosing an appropriate regularization term $\mathcal{L}_{\text{uniform}}$, leading to the third term becoming 0 via $p_\omega$. In short, Equation 6 implies that contrastive learning on distribution $\mathcal{G}$ with augmentation function $\tau$ essentially optimizes the upper-bound of supervised risk on the augmented distribution $\mathcal{G}_\tau$ resulting in a lower supervised risk. So, even without labels, we can approach the goal formulated as Equation 5 during pre-training to some extent, through modifying the main components in current GCL methods which will be discussed in Section 3.

## 3 SHIFT-ROBUST GRAPH CONTRASTIVE LEARNING

In this section, we will provide a **M**odel-**A**gnostic **R**ecipe for **I**mproving **O**OD generalization of GCL methods, dubbed MARIO. A GCL training pipeline can be typically decomposed into three components: (i) view generation, (ii) view encoding, and (iii) representation contrasting, as illustrated in Figure 1. MARIO works on the first (view generation) and the last component (representation contrasting), leaving the view encoding as an orthogonal design choice for GCL methods. Therefore it can be applied to various graph encoding models such as GCN [33], GAT [61], GraphSAGE [21] and etc.

In the remaining content, we will first analyze the drawbacks of the two components in existing GCL methods for OOD generalization and introduce our proposed recipe correspondingly in Sec. 3.1 and Sec. 3.2. Finally, we will formulate the complete training scheme for graph OOD generalization problem in Sec. 3.3. **The complete derivation and more detailed illustration of all lemmas, theorems and corollaries can be found in the Appendix.**

## 3.1 Recipe 1: Revisiting Graph Augmentation

Data augmentation plays a crucial role in the transferability and generalization ability of contrastive learning [10, 28, 58, 74]. It is proved that contrastive learning on distribution $\mathcal{G}$ with augmentation function $\tau$ essentially optimizes the supervised risk on the augmented distribution $\mathcal{G}_\tau$ instead of the original distribution $\mathcal{G}$ [74]. Consequently, if the downstream distribution $\mathcal{G}^{\text{tar}}$ is similar to training distribution $\mathcal{G}$, the encoder obtained by contrastive learning shall perform well on it. Although the alignment loss in the contrastive learning achieves certain level of generalization, the learned representation distribution lacks domain invariance since it only takes expectation over the augmentation distribution $\pi$ [74]. This limitation will hinder the OOD generalization of models [4, 74].

Our first improvement of graph contrastive learning based on graph augmentation, inspired by invariant learning [4, 74], aims to acquire domain-invariant features across $\{\mathcal{G}_\tau\}_{\tau \in \mathcal{T}}$ to address *Challenge 1*. Firstly, let us retrospect invariant risk minimization [4].

DEFINITION 1 (INVARIANT RISK MINIMIZATION, IRM). *If there is a classifier $p_{\omega^*}$ simultaneously optimal for all domains in $\mathcal{B}$, we will say that a data representation $g_\theta$ elicits an invariant predictor $p_{\omega^*} \circ g_\theta$ across a domain set $\mathcal{B}$:*

$$p_{\omega^*} \in \arg\min_{p_\omega} \mathcal{R}(p_\omega \circ g_\theta; \mathcal{G}) \text{ for all } \mathcal{G} \in \mathcal{B}, \quad (7)$$

*where $\mathcal{R}$ is the risk of the predictor $p_\omega \circ g_\theta$ measured on domain $\mathcal{G}$.*

Definition 1 yields the features that exhibit stable correlations with the target variable. It has been empirically and theoretically demonstrated that such features can enhance the generalization of models across distribution shifts in supervised learning [1, 4]. By setting $\mathcal{B}$ to the set of augmented graphs $\{\mathcal{G}_\tau\}_{\tau \in \mathcal{T}}$, this concept can be readily applied to graph contrastive learning methods with [53, 74]. The following definition of invariant alignment loss is the proposed objective for GCL-OOD problem, and we will draw the connection between Definition 1 and Definition 2 in Theorem 3.1.

DEFINITION 2 (INVARIANT ALIGNMENT LOSS). *The invariant alignment loss $\mathcal{L}_{\text{align}^*}$ of the graph encoder $g_\theta$ over the graph distribution $\mathcal{G}$ is defined as*

$$\mathcal{L}_{\text{align}^*}(g_\theta; \mathcal{G}) := \mathbb{E}_{G \in \mathcal{G}} \sup_{\tau, \tau' \in \mathcal{T}} \|g_\theta(\tau(G)) - g_\theta(\tau'(G))\|^2. \quad (8)$$

The invariant alignment loss measures the difference between two representations under the most "challenging" two augmentations, rather than the trivial expectation as in Equation 2. Intuitively, it avoids the situation where the encoder behaves extremely differently in different $\mathcal{G}_\tau$. A special case of binary classification problem is analysed in Appendix B to substantiate it. Then we will discuss why the supremum operator can solve such a dilemma.

THEOREM 3.1 (UPPER BOUND OF VARIATION ACROSS DIFFERENT DOMAINS [74]). *For two augmentation functions $\tau$ and $\tau'$, linear predictor $p$ and representation $g$, the variation across different domains is upper-bounded by*

$$\sup_{\tau, \tau' \in \mathcal{T}} |\mathcal{R}(p \circ g; \mathcal{G}_\tau) - \mathcal{R}(p \circ g; \mathcal{G}_{\tau'})| \leq c \cdot \|p\| \mathcal{L}_{\text{align}^*}(f, \mathcal{G}). \quad (9)$$

*Furthermore, fix $g$ and let $p_\tau \in \arg\min_p \mathcal{R}(p \circ g, \mathcal{G}_\tau)$. Then we have*

$$|\mathcal{R}(p_\tau \circ g; \mathcal{G}_{\tau'}) - \mathcal{R}(p_{\tau'} \circ g; \mathcal{G}_{\tau'})| \leq$$
$$2c \cdot (\|p_\tau\| + \|p_{\tau'}\|) \mathcal{L}_{\text{align}^*}(g, \mathcal{G}). \quad (10)$$

The complete deduction and the connection between contrastive loss and downstream risk $\mathcal{R}$ are in Appendix A. $\mathcal{L}_{\text{align}^*}$ replace the expectation over $\mathcal{T}$ with the supremum in $\mathcal{L}_{\text{align}}$ of Equation 2 resulting in $\mathcal{L}_{\text{align}}(g; \mathcal{G}, \pi) \leq \mathcal{L}_{\text{align}^*}(g; \mathcal{G})$ for all $g$ and $\pi$, and the augmentation function $\tau$ is randomly selected from the augmentation pool $\mathcal{T}$ based on distribution $\pi$. When $\mathcal{L}_{\text{align}^*}$ is optimized to a small value, it indicates that $\mathcal{R}(p \circ g; \mathcal{G}_\tau)$ remains consistent across different augmentation functions $\tau$, implying the optimal representation for $\mathcal{G}_\tau$ is similar to $\mathcal{G}_{\tau'}$. That is, representation with smaller $\mathcal{L}_{\text{align}^*}$ tends to elicit the same linear optimal predictors across different domains, a property lacking in the original alignment loss.

**Adversarial augmentation.** One issue with substituting $\mathcal{L}_{\text{align}}$ with $\mathcal{L}_{\text{align}^*}$ is the impracticality of estimating $\sup_{\tau, \tau' \in \mathcal{T}} \|g(\tau(G)) - g(\tau'(G))\|^2$, as it necessitates iterating over all augmentation methods. In order to find the worst case in the continuous space efficiently, we turn to the adversarial training [31, 35, 54, 56] to approximate the supermum operator:

$$\min_\theta \mathbb{E}_{(G,Y) \sim \mathcal{G}} \left[ \max_{\|\delta\|_p \leq \epsilon} L\left(g_\theta(X + \delta, A), Y\right) \right], \quad (11)$$

where the inner loop maximizes the loss to approximate the most challenging perturbation, whose strength $\|\delta\| \leq \epsilon$ is strictly controlled so that it does not change the semantic labels of the original view, *e.g.*, $\epsilon = 1e - 3$. Considering the training efficiency, in this paper, we follow and further modify the supervised graph adversarial training framework FLAG [35] to accommodate unsupervised graph contrastive learning as follows:

$$\min_\theta \mathbb{E}_{(G_\alpha, G_\beta) \sim \mathcal{G}} \left[ \max_{\|\delta\|_p \leq \epsilon} L\left(g_\theta(X_\alpha + \delta, A_\alpha), g_\theta(X_\beta, A_\beta)\right) \right]. \quad (12)$$

While not entirely new, using adversarial augmentation in graph learning models differs in our approach's objective. Prior works [30, 31] aimed to enhance model robustness against adversarial attacks, while we utilize adversarial augmentation to boost OOD generalization in GCL. Our contribution lies in providing theoretical justifications and deeper insights into the benefits of adversarial augmentation for OOD generalization within GCL.

## 3.2 Recipe 2: Revisiting Representations Contrasting

The vanilla contrastive loss like Equation 2 aims to maximize the lower bound of the mutual information between positive pairs. However, there exists some redundant information (*i.e.*, conditional mutual information) that can impede the generalization of graph contrastive learning. Our objective is to learn minimal sufficient representation related to downstream task which can effectively mitigate overfitting and demonstrate robustness against distribution shifts. In this subsection, we introduce a recipe for representation contrasting to improve the generalization of GCL methods, motivated by the principle of information bottleneck (IB) [59] to solve *Challenge 2*. In short, we refer to the modified contrastive loss to get rid of supervision signals as well as learning generalized representations to assist in addressing *Challenge 1*. (the IB constraint

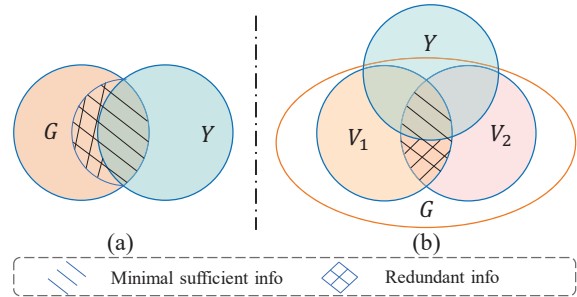

Minimal sufficient info ◇◇◇ Redundant info

**Figure 2: Venn diagram of mutual information and conditional mutual information**

in conjunction with invariance principle effectively addresses key failures linked to invariant features [1, 36].)

DEFINITION 3 (INFORMATION BOTTLENECK, IB). *Let $X, Z, Y$ represent random variables of inputs, embeddings, and labels respectively. The formulation of information bottleneck's training objective is*

$$\arg\max_\theta R_{IB}(\theta) = I_\theta(Z; Y) - \beta I_\theta(Z; X), \tag{13}$$

*where $I_\theta$ represents mutual information estimator with parameters $\theta$, and $\beta > 0$ controls the trade-off between compression and the downstream task performance (larger $\beta$ leads to lower compression rate but high MI between the embedding $Z$ and the label $Y$).*

The IB principle [3, 59] aims to learn a minimal sufficient representation for the given task. It achieves this by maximizing the mutual information between the representation and the target (*sufficiency*) while constraining the mutual information between the representation and the input data (*minimality*), as shown in Figure 2a. This learning paradigm helps combat overfitting and enhances resilience against distribution shifts [1, 36, 65].

Motivated by this principle, we modify the vanilla contrastive loss [73, 77, 78] as Equation 19. Current graph contrastive learning methods aim to maximize mutual information between positive pairs, as depicted in Figure 2b. However, in scenarios with available training labels, some information in the vanilla contrastive loss becomes redundant. In Figure 2b, $V_1$ and $V_2$ represent two augmented views from the same sample $\mathcal{G}$, and $U$ and $V$ represent their respective representations. Eliminating this redundant information aligns with the IB principle. To describe this redundancy more precisely, we introduce conditional mutual information (CMI).

DEFINITION 4 (CONDITIONAL MUTUAL INFORMATION, CMI). *The conditional mutual information $I(U; V \mid Y)$ measures the expected value of mutual information between $U$ and $V$ given $Y$ which can be formulated as*

$$I(U; V \mid Y) := \mathbb{E}_{y \sim Y} \left[ D_{KL} \left( P_{U,V|Y=y} \| P_{U|Y=y} P_{V|Y=y} \right) \right] \\ = \int_{\mathcal{Y}} D_{KL} \left( P_{U,V|Y} \| P_{U|Y} P_{V|Y} \right) dP_Y. \tag{14}$$

To reduce the redundant information and hence improve the OOD generalization ability, we need to minimize the CMI between two views $U$ and $V$. However, it is intractable to estimate the equation above. In this work, we appeal to mutual information estimators (*e.g.*, Donsker-Varadhan estimator [6, 14], Jensen-Shannon estimator [6, 46], InfoNCE [20, 47]) to estimate the conditional

mutual information. Taking InfoNCE [47] as an example, the CMI objective can be approximated as

$$\mathcal{L}_{CMI}(U, V) = -\mathbb{E}_{y \sim P_Y} \Big[ \mathbb{E}_{u,v \sim P_{U,V|y}} \left[ \text{sim}(u, v) \right] \\ + \mathbb{E}_{u \sim P_{U|y}} \log \mathbb{E}_{v^- \sim P_{V|y}} \left[ e^{\text{sim}(u,v^-)} \right] \Big] \tag{15}$$

where $\text{sim}(x, y)$ is the cosine similarity function, the positive pairs are drawn from the conditional joint distribution, and negative pairs are drawn from the product of conditional marginal distribution. In short, we first sample $y \sim Y$, and then we sample positive and negative pairs from $P_{U,V|y}$ and $P_{U|y} P_{V|y}$. The negative format of Equation 2 is a lower bound of the conditional mutual information $I(U; V \mid Y)$. The proof can be found in Appendix D.

**Online Clustering.** The main challenge of applying the above approximation to our unsupervised pre-training is the lack of labels $Y$. To address this issue, we utilize online clustering techniques to obtain pseudo labels. These pseudo labels are iteratively refined during training, ensuring an increase in their mutual information with the ground-truth labels [39]. To integrate clustering into our pre-text task, we employ strategies similar to [9, 39]. We will initialize learnable prototypes $c_i$ for each cluster $i$ and matrix $C = [c_1, c_2, \cdots, c_K]$ collects all column prototype vectors. For clustering, we can simply calculate the similarity between $K$ prototypes and node representations $u_i$ and $v_i$ for node $i$:

$$p_{u_i}(\hat{y} \mid u_i) = \text{softmax}\left( C^T \cdot u_i \right), \quad q_{v_i}(\hat{y} \mid v_i) = \text{softmax}\left( C^T \cdot v_i \right), \tag{16}$$

where the prototypes $C$ are updated by solving the problem of swapped prediction [9]:

$$\mathcal{L}_{clu}(U, V) = \sum_i^B \left[ \ell(p_{u_i}, q_{v_i}) + \ell(q_{u_i}, p_{v_i}) \right], \\ \text{where} \quad \ell(p_{u_i}, q_{v_i}) = -\sum_k q_{v_i}^{(k)} \log p_{u_i}^{(k)}. \tag{17}$$

The clustering loss focuses on contrasting nodes by comparing cluster assignments rather than their representations. However, it may lead to a trivial solution where all samples are in one cluster. To prevent this, we introduce a constraint for equal prototype assignment partition [9]. Details can be found in Appendix E.

For stable training, we use bi-level optimization [41] for updating the encoder and prototypes (more details in Section 3.3). With these prototypes, we can infer the pseudo labels of node representations:

$$\hat{Y} = \arg\max C^T U. \tag{18}$$

Hence our final shift-robust contrastive loss can be formulated as

$$\min_{g_\theta} \mathcal{L}_{rob} = \arg\max I_\theta(U; V) - \gamma I_\theta(U; V \mid \hat{Y}) \\ = \arg\min_{g_\theta} \mathcal{L}_{MI} - \gamma \mathcal{L}_{CMI}, \tag{19}$$

where $I_\theta(U; V), I_\theta(U; V \mid \hat{Y})$ can be instantiated as Equation 2 and Equation 19 respectively; $\gamma \geq 0$ controls the trade-off between compression and pre-text task's performance similar in Equation 13. Intuitively, if the positive pairs have already shared the same semantic labels in the feature space (*i.e.*, belong to the same cluster), the objective will reduce their shared information to avoid learning redundant information and overfitting [3, 65] during training, which will bring performance gain in OOD generalization.

## 3.3 Model Training

Our shift-robust contrastive loss is formulated as Equation 19 which involves maximizing the mutual information and minimizing the conditional mutual information at the same time. As mentioned in Section 3.2, bi-level optimization [41] is used for updating the prototypes $C$ and other parameters in Equation 19:

$$g^{k+1} = \arg\min_g \mathcal{L}_{\text{rob}}(g^k, C^{k+1}; \mathcal{G}, \mathcal{T}),$$
$$C^{k+1} = \arg\min_C \mathcal{L}_{\text{clu}}(g^k, C; \mathcal{G}, \mathcal{T}), \tag{20}$$

where the parameter of the graph encoder $g_\theta$ is omitted for uncluttered notations. Concretely, we first fix the encoder $g$ and update the prototypes with $l$ steps of SGD to approximate the arg min optimization, and then with the near-optimal prototypes $C$, we update the parameters of the encoder for 1 step of SGD. With the adversarial augmentation, our final optimization objective $\mathcal{L}_{\text{rob}}$ for the graph encoder is replaced by:

$$\min_g \mathbb{E}_{(G_\alpha, G_\beta) \sim \mathcal{G}} \left[ \max_{\|\delta\|_p \leq \epsilon} \mathcal{L}_{\text{rob}} \left( g(X_\alpha + \delta, A_\alpha), g(X_\beta, A_\beta), C \right) \right]. \tag{21}$$

The combination of Invariance principle (Recipe 1) and Information Bottleneck principle (Recipe 2) further enhances OOD generalization in learned representations. Because invariance with IB constraint helps solve key failures linked to invariant features [1, 36]. The ablation study in Sec. 4.3 confirms this synergy in unsupervised context, surpassing the performance of individual components. Furthermore, the efficiency of our method is comparable to GRACE [78]. More details and Algorithm can be found in Appendix F.

## 4 EXPERIMENTS

In this section, we first introduce the experimental setup including datasets, training, and evaluation protocol in Sec. 4.1 and 4.2. We then perform an ablation study to demonstrate the effectiveness of each proposed component in Sec. 4.3. Lastly, we analyze the impact of important hyper-parameters in Appendix 4.4. It is important to note that we focus on node-level tasks (*e.g.*, node classification) in this work. As for graph-level tasks, we leave them as our future work, while some simple experiments are also provided in Appendix H.1. Additionally, we integrate our method with various encoding models, showcasing the model-agnostic nature of our recipe in Appendix H.3. And we provide some qualitative results such as feature visualization in Appendix H.5.

## 4.1 Datasets

There exist some benchmarks for evaluating graph out-of-distribution generalization [13, 19, 29]. Among them, GOOD [19] is the most representative and comprehensive one. It curates more diverse graph datasets with diverse tasks, including single/multi-task graph classification, graph regression, and node classification involving more distribution shifts, *i.e.*, concept shift (supp $(P_{\text{train}}(X)) \approx$ supp $(P_{\text{test}}(X))$ and $P_{\text{train}}(Y|X) \neq P_{\text{test}}(Y|X)$) [17] and covariate shift ($P_{\text{train}}(Y|X) = P_{\text{test}}(Y|X)$ and $P_{\text{train}}(X) \neq P_{\text{text}}(X)$) [55]. Hence in this work, we follow the evaluation protocol proposed in [19]. Furthermore, we validate the effectiveness of our method in the datasets (*i.e.*, Amazon-Photo, Elliptic) that are

used in EERM [64]. The statistics and detailed introduction to these datasets can be found in Table 4 and Appendix G.2.

## 4.2 Unsupervised Representation Learning

*4.2.1 Transductive Setting.* In this subsection, we validate our proposed algorithm in the transductive setting, where test nodes participate in message passing during training, following [19].

**Baselines:** We conduct experiments with 12 baselines from three categories: (i) supervised methods, including empirical risk minimization (**ERM**) [60], **FLAG** [35], invariant risk minimization (**IRM**) [4], and graph OOD method **EERM** [64]; (ii) self-supervised generative methods including **GAE** [32], **VGAE** [32], **Graph-MAE** [25]; (iii) self-supervised contrastive methods: **DGI** [62], **MV-GRL** [22], **GRACE** [78], **RoSA** [77], **BGRL** [57], **COSTA** [73], **SwAV** [9]. The descriptions of baselines are in Appendix G.1.

**Experimental setup:** To ensure a fair comparison, we use the same model configuration for all methods, following [19], and perform grid search to find optimal hyperparameters like learning rate and epochs. More details can be found in Appendix G.3.

**Analysis:** Based on the experimental results listed in Table 1 and 2, we can draw the following conclusions: firstly, we find strong self-supervised methods (*e.g.*, GRACE, BGRL, COSTA) are more robust to distribution shifts compared to supervised methods. Secondly, we find the methods designed for OOD generalization (*i.e.*, IRM and graph OOD generalization (*i.e.*, EERM) do not attain superior performance than ERM on most of the datasets. This phenomenon is also observed in [2, 19, 52], showcasing the challenge of achieving invariant prediction in non-Euclidean graph settings.

Furthermore, our method surpasses other SOTA self-supervised methods by a considerable margin on all OOD test sets while achieving comparable performance on in-distribution test sets. For instance, on small datasets such as GOOD-CBAS, our method outperforms GRACE[5] by over 2% absolute accuracy on OOD test set. On larger datasets like GOOD-Cora and GOOD-Twitch, our method consistently outperforms others, with over 7% absolute accuracy improvement on GOOD-Twitch's OOD test set under covariate shift. These statistics prove the effectiveness of our design.

*4.2.2 Inductive Setting.* In this subsection, we conduct experiments under inductive settings, where test nodes are kept unseen during training. This setting is more suitable for domain generalization.

**Baselines:** For GOOD-WebKB and GOOD-CBAS, we adopt ERM, IRM, GraphMAE, and GRACE as baselines. For Amazon-Photo and Elliptic datasets, we select ERM, EERM, and GRACE as our baselines. As for the experimental setup, please refer to Appendix G.3.

**Experimental setup:** We conduct experiments on Amazon-Photo dataset [68] and Elliptic [48] dataset in this subsection. These datasets consist of many snapshots (training data and testing data use different snapshots) which are naturally inductive. As for the inductive setting on GOOD-WebKB and GOOD-CBAS datasets, the results and analysis can be found in Appendix H.2.

**Analysis:** According to Figure 3,4,8, we can draw following conclusions: firstly, based on Figure 3, it is evident that our method outperforms other representative supervised and self-supervised methods on all test graphs (T1~T8). This superiority is reflected in

---

[5]We compare MARIO with GRACE [78] since it is built upon the latter method according to our proposed recipe.

**Table 1: Experimental results of all methods under concept shift. The bold font means the top-1 performance and the underline represents the second performance across the unsupervised methods. 'ID' represents in-distribution test performance and 'OOD' means out-of-distribution test performance. (OOM: out-of-memory on a GPU with 24GB memory)**

| concept shift | GOOD-Cora | | | | GOOD-CBAS | | GOOD-Twitch | | GOOD-WebKB | |
| | word | | degree | | color | | language | | university | |
| | ID | OOD | ID | OOD | ID | OOD | ID | OOD | ID | OOD |
|---|---|---|---|---|---|---|---|---|---|---|
| ERM [60] | 66.38±0.45 | 64.44±0.18 | 68.60±0.40 | 60.76±0.34 | 89.79±1.39 | 83.43±1.19 | 80.80±1.00 | 56.92±0.92 | 62.67±1.53 | 26.33±1.09 |
| FLAG [35] | 66.84±0.51 | 65.15±0.17 | 68.48±0.95 | 61.25±0.76 | 88.43±2.32 | 82.14±0.55 | 80.93±0.84 | 57.24±0.64 | 60.33±1.64 | 25.78±1.39 |
| IRM [4] | 66.42±0.41 | 64.29±0.31 | 68.57±0.35 | 61.45±0.24 | 89.64±1.21 | 82.29±1.14 | 78.87±1.04 | 59.30±1.79 | 62.67±1.10 | 26.88±1.42 |
| EERM [64] | 65.10±0.44 | 62.45±0.19 | 66.95±0.44 | 56.58±0.25 | 79.07±2.12 | 64.50±1.01 | OOM | OOM | 62.50±2.01 | 28.07±3.23 |
| GAE [32] | 60.65±0.89 | 58.00±0.55 | 62.59±1.11 | 53.44±0.80 | 75.28±1.36 | 68.07±2.05 | 81.25±0.81 | 51.51±1.05 | 62.17±3.34 | 25.78±1.85 |
| VGAE [32] | 63.19±0.53 | 60.35±0.47 | 61.65±0.66 | 54.28±0.28 | 76.50±0.50 | 59.07±0.56 | 80.46±0.53 | 55.56±4.53 | 62.50±2.38 | 24.40±2.57 |
| GraphMAE [25] | 66.44±0.46 | 64.87±0.30 | 67.95±0.46 | 59.41±0.39 | 89.14±0.89 | 82.93±0.93 | 80.05±0.64 | 59.38±1.49 | 61.83±3.37 | 29.27±2.15 |
| DGI [62] | 63.33±0.56 | 60.71±0.49 | 65.93±1.02 | 55.83±0.53 | 91.22±1.47 | 85.00±1.66 | 80.05±0.87 | 59.16±1.88 | 61.83±2.83 | 28.63±1.92 |
| MVGRL [22] | OOM | OOM | OOM | OOM | 88.57±1.15 | 76.50±1.17 | OOM | OOM | 62.00±3.79 | 28.26±4.20 |
| GRACE [78] | 65.61±0.61 | 63.92±0.44 | 68.59±0.35 | 60.15±0.45 | 92.00±1.39 | 88.64±0.67 | 83.43±0.63 | 60.45±1.46 | 64.00±3.43 | 34.86±3.43 |
| RoSA [77] | 64.06±0.67 | 62.44±0.39 | 67.07±0.65 | 57.68±0.44 | 90.78±2.27 | 85.93±2.14 | 82.39±0.42 | 57.45±2.16 | 64.17±4.10 | 32.20±2.15 |
| BGRL [57] | 65.18±0.43 | 63.43±0.45 | 66.83±0.80 | 59.63±0.38 | 92.36±1.16 | 87.14±1.60 | 82.52±0.60 | 55.48±1.48 | 63.67±2.33 | 31.47±3.43 |
| G-BT [7] | 64.85±0.59 | 63.29±0.26 | 67.23±0.80 | 56.89±0.46 | 92.50±1.07 | 88.36±1.24 | 83.28±0.27 | 58.82±1.75 | 63.50±2.03 | 31.74±2.50 |
| COSTA [73] | 65.05±0.80 | 62.37±0.45 | 66.76±0.87 | 55.73±0.36 | 93.50±2.62 | 89.29±3.11 | 83.15±0.30 | 55.03±3.22 | 61.66±2.58 | 32.39±2.13 |
| SwAV [9] | 62.22±0.53 | 59.79±0.53 | 64.65±0.94 | 55.06±0.39 | 89.00±0.79 | 81.72±0.66 | 83.32±0.15 | 59.69±1.97 | 65.17±3.76 | 29.36±2.01 |
| MARIO (ours) | 67.11±0.46 | 65.28±0.34 | 68.46±0.40 | 61.30±0.28 | 94.36±1.21 | 91.28±1.10 | 82.31±0.54 | 63.33±1.72 | 65.67±2.81 | 37.15±2.37 |

**Table 2: Experimental results of all methods under covariate shift.**

| covariate shift | GOOD-Cora | | | | GOOD-CBAS | | GOOD-Twitch | | GOOD-WebKB | |
| | word | | degree | | color | | language | | university | |
| | ID | OOD | ID | OOD | ID | OOD | ID | OOD | ID | OOD |
|---|---|---|---|---|---|---|---|---|---|---|
| ERM [60] | 70.50±0.41 | 64.69±0.33 | 72.46±0.49 | 55.53±0.50 | 92.00±3.08 | 77.57±1.29 | 70.98±0.41 | 49.35±5.09 | 39.34±1.79 | 14.52±3.14 |
| FLAG [35] | 70.82±0.45 | 64.98±0.22 | 72.88±0.31 | 56.25±0.33 | 92.43±2.12 | 79.00±1.70 | 70.69±0.59 | 46.87±1.86 | 40.65±1.77 | 14.68±3.04 |
| IRM [4] | 70.48±0.26 | 64.53±0.57 | 71.98±0.34 | 53.72±0.46 | 90.86±2.41 | 78.86±1.67 | 69.81±0.95 | 49.11±2.82 | 38.52±3.30 | 13.97±2.80 |
| EERM [64] | OOM | OOM | OOM | OOM | 65.00±2.57 | 57.43±3.60 | OOM | OOM | 46.07±4.55 | 27.40±7.65 |
| GAE [32] | 56.63±0.79 | 48.93±0.93 | 66.30±0.88 | 34.01±0.87 | 73.00±2.16 | 60.80±3.01 | 67.24±1.23 | 47.65±2.49 | 45.08±6.32 | 28.02±6.29 |
| VGAE [32] | 62.02±0.66 | 54.12±0.86 | 69.41±0.57 | 44.20±1.29 | 62.29±2.04 | 63.29±1.11 | 66.99±1.43 | 50.48±4.58 | 50.20±6.69 | 20.87±6.69 |
| GraphMAE [25] | 68.14±0.43 | 64.00±0.33 | 73.36±0.56 | 53.75±0.55 | 67.28±3.03 | 67.28±1.49 | 68.84±1.20 | 48.02±2.79 | 48.03±4.34 | 30.00±8.09 |
| DGI [62] | 60.85±0.75 | 57.03±0.67 | 68.97±0.41 | 41.75±0.88 | 69.57±4.09 | 59.71±3.43 | 68.43±1.05 | 44.83±1.61 | 48.52±5.04 | 21.11±7.50 |
| MVGRL [22] | OOM | OOM | OOM | OOM | 65.00±1.94 | 64.15±0.77 | OOM | OOM | 54.10±5.39 | 16.59±6.51 |
| GRACE [78] | 68.77±0.33 | 64.21±0.41 | 72.69±0.34 | 56.10±0.63 | 93.57±1.83 | 89.29±3.40 | 71.12±0.87 | 46.21±1.54 | 49.67±5.82 | 28.10±4.68 |
| RoSA [77] | 68.19±0.56 | 62.48±0.61 | 71.04±0.62 | 52.72±0.79 | 84.71±4.14 | 79.14±3.51 | 70.58±0.36 | 45.83±1.72 | 52.30±4.24 | 34.24±7.92 |
| BGRL [57] | 67.23±0.43 | 61.33±0.36 | 72.11±0.39 | 49.15±0.73 | 89.00±2.56 | 79.86±3.29 | 71.43±0.53 | 43.86±0.94 | 51.80±5.55 | 30.32±7.61 |
| G-BT [7] | 67.72±0.37 | 63.34±0.52 | 69.89±0.60 | 54.18±0.55 | 94.00±1.24 | 91.29±2.67 | 71.25±0.75 | 46.36±1.60 | 53.77±4.80 | 25.48±8.81 |
| COSTA [73] | 65.28±0.60 | 60.33±0.53 | 70.65±0.62 | 54.03±0.28 | 92.29±1.59 | 82.71±2.74 | 69.29±1.37 | 49.07±2.13 | 50.49±3.01 | 29.84±4.75 |
| SwAV [9] | 63.29±1.01 | 56.98±0.94 | 70.27±0.73 | 43.00±0.52 | 89.57±1.12 | 81.43±1.69 | 69.19±0.93 | 49.37±2.96 | 49.84±4.82 | 30.55±6.72 |
| MARIO(ours) | 69.99±0.54 | 65.06±0.34 | 72.73±0.43 | 57.73±0.45 | 95.43±1.40 | 95.00±2.41 | 68.31±0.78 | 57.37±1.37 | 53.94±3.23 | 35.24±4.98 |

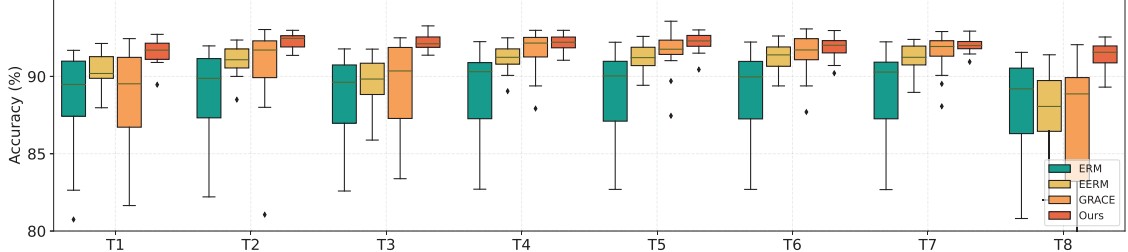

**Figure 3: Results on Amazon-photo with artificial distribution shifts. T1-T8 are distinct test graphs created by environment IDs.**

the larger median value of our method compared to others. For instance, MARIO achieves over a 3% absolute improvement compared to ERM in terms of the mean value of eight median values.

Secondly, from the results presented in Figure 4, we can observe that our method averagely harvests 10.9% absolute improvement over GRACE and 12.5% absolute improvement over EERM in terms of F1 scores on Elliptic dataset. This demonstrates the effectiveness of our method in handling distribution shifts and improving performance compared to existing approaches. It is worth noting that GRACE's performance worsens over time, indicating its inability to

**Table 3: Ablation studies for MARIO by masking each component.**

| concept shift | GOOD-Cora | | | | GOOD-CBAS | | GOOD-Twitch | | GOOD-WebKB | |
| --- | --- | --- | --- | --- | --- | --- | --- | --- | --- | --- |
| | word | | degree | | color | | language | | university | |
| | ID | OOD | ID | OOD | ID | OOD | ID | OOD | ID | OOD |
| MARIO | **67.11±0.46** | **65.28±0.34** | **68.46±0.40** | **61.30±0.28** | **94.36±1.21** | **91.28±1.10** | 82.31±0.54 | **63.33±1.72** | **65.67±2.81** | **37.15±2.37** |
| MARIO(w/o ad) | 66.23±0.53 | 64.02±0.18 | 67.88±0.38 | 60.46±0.29 | 93.21±1.25 | 90.29±0.91 | 82.42±0.73 | 60.50±1.02 | 64.83±2.83 | 36.51±3.25 |
| MARIO(w/o cmi) | 65.32±0.60 | 63.51±0.32 | 68.14±0.32 | 61.19±0.34 | 94.15±1.23 | 90.57±1.96 | **82.51±0.56** | 61.41±2.63 | 64.50±4.35 | 35.78±2.53 |
| MARIO(w/o cmi, ad) | 64.67±0.55 | 63.11±0.32 | 67.95±0.65 | 60.01±0.57 | 93.36±1.66 | 89.64±1.73 | 81.90±0.75 | 60.12±1.60 | 64.17±3.67 | 34.13±2.38 |

handle distribution shifts effectively. In contrast, our method consistently achieves better F1 scores, except for T9, which is caused by the dark market shutdown occurred after T7 [48].

Overall, the observations we have made provide strong evidence of the great capacity of our method for handling distribution shifts, validating its effectiveness and potential for real-world applications.

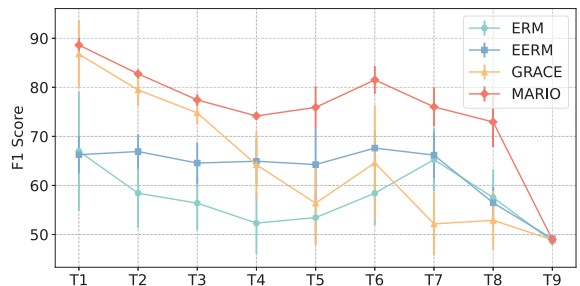

**Figure 4: Experiment results on Elliptic dataset with label distribution shifts. T1~T9 denote different groups of test graph snapshots according to the chronological order.**

## 4.3 Ablation Studies

Table 3 provides a detailed analysis of the effect of each component according to our proposed recipe for improving OOD generalization in graph contrastive learning. We examine the different variants of our method and their impact on performance. Specifically, MARIO (w/o ad) represents MARIO without adversarial augmentation. MARIO (w/o cmi) denotes we only maximize the mutual information between positive pairs without considering conditional mutual information. MARIO (w/o cmi, ad) means a vanilla graph contrastive method that is similar to GRACE.

From Table 3, we observe that MARIO (w/o cmi) significantly underperforms MARIO on OOD test sets, emphasizing the importance of minimizing redundant information for improving OOD generalization in GCL methods. Adversarial augmentation also contributes to improved OOD generalization by approximating a supremum operator, enabling the learning of more invariant features, as discussed in Section 3.1. In summary, the analysis of these variants confirms the effectiveness of the proposed enhancements in data augmentation and contrastive loss. Each component enhances performance, and their combination creates a stronger self-supervised graph learner with improved graph OOD generalization.

## 4.4 Sensitivity Analysis

In this section, we perform sensitivity analysis on two crucial hyperparameters of our method using the GOOD-WebKB dataset with concept shift. We explore different values for the coefficient $\gamma$ in

Equation 19 and the number of prototypes $|C|$ in Equation 16. The analysis reveals that the best OOD test accuracy is achieved with $\gamma = 0.1$ and $|C|$ set to 100 or 200. Both higher and lower values of $\gamma$ and $|C|$ result in suboptimal performance. These findings emphasize the importance of selecting appropriate hyper-parameter values to balance compression levels and pseudo label counts which are also found in DIB [3], leading to improved graph OOD generalization. Based on the sensitivity analysis, we determined that setting $\gamma = 0.1$ and $|C| = 100$ on most datasets, as these hyperparameter values optimize graph OOD generalization while maintaining a reasonable compression level and pseudo label count.

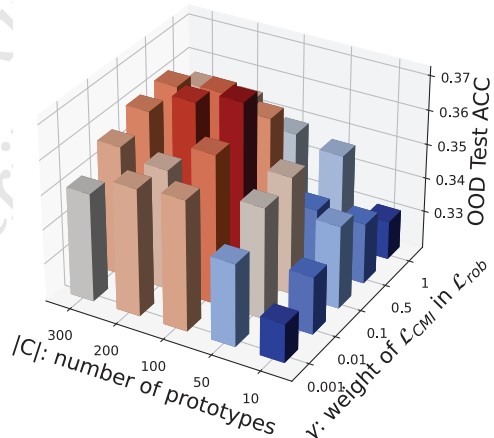

**Figure 5: Sensitivity Analysis on CMI coefficient and the number of prototypes.**

## 5 CONCLUSION

In this work, we propose a model-agnostic recipe called MARIO (Model-Agnostic Recipe for Improving OOD Generalization) to address the challenges of distribution shifts in graph contrastive learning. Specifically, this recipe mainly aims to address the drawbacks of the main components (*i.e.*, view generation and representation contrasting) in graph contrastive learning while facing distribution shifts motivated by invariant learning and information bottleneck principles. To the best of our knowledge, this is the first work that investigates the OOD generalization problem of graph contrastive learning specifically for node-level tasks. We conduct substantial experiments to show the superiority of our method on various real-world datasets with diverse distribution shifts. This research contributes to bridging the gap in understanding and addressing distribution shifts in graph contrastive learning, providing valuable insights for future research in this area.

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

## A PROOFS IN SECTION 2

In this section, we will illustrate some notations mentioned in Equation 6 firstly, and then we will provide a proof of Equation 6.

DEFINITION 5 (($\sigma, \delta$)-augmentation[28]). *Let $C_k \subseteq \mathcal{G}$ denote the set of all points in class $k$. A graph augmentation set $\mathcal{T}$ can be referred to as a ($\sigma, \delta$)-augmentation on $\mathcal{G}$, where $\sigma \in (0, 1]$ and $\delta > 0$. This is the case if, for every $k \in [K]$, there exists a subset $C_k^0 \subseteq C_k$ such that the following conditions hold:*

$$
\begin{aligned}
& P_{G \sim \mathcal{G}} \left( G \in C_k^0 \right) \geq \sigma P_{G \sim \mathcal{G}} \left( G \in C_k \right), \\
& and \sup_{G_1, G_2 \in C_k^0} d_A \left( G_1, G_2 \right) \leq \delta,
\end{aligned}
\tag{22}
$$

*where $d_{\mathcal{A}} \left( G_1, G_2 \right) := \inf_{\tau_1, \tau_2 \in \mathcal{T}} d \left( \tau_1 \left( G \right), \tau_2 \left( G \right) \right)$ for some distance $d(\cdot, \cdot)$.*

This definition quantifies the concentration of augmented data. An augmentation set with a smaller value of $\delta$ and a larger value of $\sigma$ results in a more clustered arrangement of the original data. In other words, samples from the same class are closer to each other after augmentation. Consequently, one can anticipate that the learned representation $g_\theta$ will exhibit improved cluster performance. This principle was proposed in [28] and modified to a more practical scenario by [74].

**Proof of Equation 6.** For simplicity, we omit the notations $\mathcal{G}$ and $\pi$ here and omit the parameters $\theta, \omega$ of $g_\theta$ and $p_\omega$, and use $G_1$ to denote the augmented data, use $t_k$ to represent $\mathcal{G}_\pi(C_k)$ and use $\mu_k$ to denote $\mu_k(g; \mathcal{G}_\pi)$. Based on Theorem 2, Lemma B.1 in [28], and Appendix B in [74], we have

$$
\mathbb{E}_{G \in C_k} \|g(G_1) - \mu_k\| \leq c \sqrt{\frac{1}{t_k} \mathcal{L}_{\text{align}}^{\frac{1}{4}}(g)} + \zeta(\sigma, \delta)
\tag{23}
$$

for some constant $c$, where

$$
\zeta(\sigma, \delta) := 4 \left( 1 - \sigma \left( 1 - \frac{L\delta}{4} \right) \right)
\tag{24}
$$

We can find $\zeta$ is decreasing with $\sigma$ and increasing with $\delta$. So, we can obtain:

$$
\begin{aligned}
& c \left( \sum_{k=1}^{K} \sqrt{t_k} \right) \mathcal{L}_{\text{align}}^{\frac{1}{4}}(f) + \zeta(\sigma, \delta) \\
& \geq \sum_{k=1}^{K} t_k \mathbb{E}_{G \in C_k} \|g(G_1) - \mu_k\| \\
& \geq \sum_{k=1}^{K} \frac{t_k}{\|p\|} \mathbb{E}_{G \in C_k} \mathbb{E}_{G_1 \in \tau(G)} \|p \circ g(G_1) - p \circ \mu_k\| \\
& \geq \sum_{k=1}^{K} \frac{t_k}{\|p\|} \mathbb{E}_{G \in C_k} \mathbb{E}_{G_1 \in \tau(G)} \|p \circ g(G_1) - e_k\| \\
& \quad - \frac{1}{\|p\|} \sum_{k=1}^{K} t_k \|e_k - p \circ \mu_k\| \\
& = \frac{1}{\|p\|} \mathcal{R}(p \circ g) - \frac{1}{\|p\|} \sum_{k=1}^{K} t_k \|e_k - p \circ \mu_k\|
\end{aligned}
\tag{25}
$$

for all linear layer $p \in \mathcal{R}^{K \times d_1}$. Therefore, we obtain

$$
\begin{aligned}
\mathcal{R}(p \circ g) & \leq c\|p\| \mathcal{L}_{\text{align}}^{\frac{1}{4}} \sum_{k=1}^{K} \sqrt{t_k} + \|p\| \zeta(\sigma, \delta) \\
& \quad + \sum_{k=1}^{K} t_k \|e_k - p \circ \mu_k\| \\
& \leq c\|p\| \sqrt{K} \mathcal{L}_{\text{align}}^{\frac{1}{4}} + \|p\| \zeta(\sigma, \delta) \\
& \quad + \sum_{k=1}^{K} t_k \|e_k - p \circ \mu_k\|.
\end{aligned}
\tag{26}
$$

## B SPECIAL CASE

The case adopted from [74] is used to prove the encoders learned through contrastive learning could behave extremely differently in different $\mathcal{G}_\tau$.

PROPOSITION B.1. *Consider a binary classification problem with data $(X_1, X_2) \sim \mathcal{N}(0, I_2)$. If $X_1 \geq 0$, the label $Y = 1$, and the data augmentation is to multiply $X_2$ by standard normal noise:*

$$
\begin{aligned}
\tau_\theta(X) & = (X_1, \theta \cdot X_2) \\
\theta & \sim \mathcal{N}(0, 1)
\end{aligned}
\tag{27}
$$

*The transformation-induced domain set is $\mathcal{B} = \{\mathcal{G}_c : \mathcal{G}_c = (X_1, c \cdot X_2) \text{ for } c \in \mathbb{R}\}$. Considering the 0-1 loss, $\forall \varepsilon \geq 0$, there holds representation $g$ and two domains $\mathcal{G}_c$ and $\mathcal{G}_{c'}$ such that*

$$
\mathcal{L}_{align} (g; \mathcal{G}, \pi) < \varepsilon
\tag{28}
$$

*but $g$ behaves extremely differently in different domains $\mathcal{G}_c$ and $\mathcal{G}_{c'}$:*

$$
|\mathcal{R}(g; \mathcal{G}_c) - \mathcal{R}(g; \mathcal{G}_{c'})| \geq \frac{1}{4}
\tag{29}
$$

This instance[6] illustrates that the obtained representation with small contrastive loss will still exhibit significantly varied performance over different augmentation-induced domains. The underlying idea behind this example lies in achieving a small $\mathcal{L}_{align}$ by aligning different augmentation-induced domains in an average sense, rather than a uniform one. Consequently, the representation might still encounter large alignment losses on certain infrequently chosen augmented domains.

*Proof.* For $\varepsilon \geq 0$, let $t = \sqrt{\varepsilon}/2$ and $g(x_1, x_2) = x1 + tx_2$. Then, the alignment loss of $g$ satisfies:

$$
\mathcal{L}_{\text{align}} (g; \mathcal{G}, \pi) = t^2 \mathbb{E} X_2^2 \mathbb{E}_{(\theta_1, \theta_2) \sim \mathcal{N}(0,1)^2} (\theta_1 - \theta_2)^2 = 2t^2 < \varepsilon.
\tag{30}
$$

Set $c$ as 0 and $c'$ as $1/t$, it is obviously that:

$$
\mathcal{R}(g; \mathcal{G}_c) = 0
\tag{31}
$$

but

$$
\begin{aligned}
& \mathcal{R}(g; \mathcal{G}_{c'}) = \\
& P(X_1 < 0, X_1 + X_2 \geq 0) + P(X_1 \geq 0, X_1 + X_2 \leq 0) = \frac{1}{4}
\end{aligned}
\tag{32}
$$

---

[6]For simplicity, we assume the adjacent matrix is an identity matrix here.

## C PROOF OF THEOREM 3.1

*Proof of Theorem 3.1:*

$$\mathcal{R}\left(p \circ g; \mathcal{G}_\tau\right) - \mathcal{R}\left(p \circ g; \mathcal{G}_{\tau'}\right)$$

$$= \mathbb{E}_{(G,Y) \sim \mathcal{G}} \left(\left|p \circ g(\tau(G)) - Y\right|^2 - \left|p \circ g\left(\tau'(G)\right) - Y\right|^2\right)$$

$$= \mathbb{E}_{(G,Y) \sim \mathcal{G}} \left(p \circ g(\tau(G)) - p \circ g\left(\tau'(G)\right)\right)\left((p \circ g(\tau(G)) + \right.$$

$$\left. p \circ g\left(\tau'(G)\right)\right) + 2Y\right)$$

$$\leq c \mathbb{E}_{(G,Y) \sim \mathcal{G}} \left\| p \circ g(\tau(G)) - p \circ g\left(\tau'(G)\right)\right\|$$

$$\leq c \|p\| \mathbb{E}_{(G,Y) \sim \mathcal{G}} \left\| g(\tau(G)) - g\left(\tau'(G)\right)\right\|$$

$$\leq c \|p\| \mathcal{L}_{\text{align}^*}(g)$$

$$\tag{33}$$

## D PROOF OF CMI LOWER BOUND

In this section, we provide a theoretical justification of why Equation 2 is a lower bound of CMI. And some justifications are borrowed from [42, 45]. Firstly, we present the following lemmas which will be used.

### D.1 Fundamental Lemmas

LEMMA D.1. *Let $U$ and $V$ be two random variables whose sample spaces are $\mathcal{U}$ and $\mathcal{V}$, $f : (\mathcal{U} \times \mathcal{V}) \to \mathbb{R}$ be a mapping function, and $\mathbb{P}$ and $\mathbb{Q}$ be the probability measures on $\mathcal{U} \times \mathcal{V}$, we can obtain:*

$$D_{\text{KL}}(\mathbb{P}\|\mathbb{Q}) = \sup_f \mathbb{E}_{(u,v) \sim \mathbb{P}}[f(u,v)] - \mathbb{E}_{(u,v) \sim \mathbb{Q}} \left[e^{f(u,v)}\right] + 1 \quad (34)$$

*Proof.* The second-order functional derivative of the above function is $-e^{f(u,v)} \cdot d\mathbb{Q}$. This negative term means Equation 34 has a supreme value. Through setting the first-order functional derivative as zero $d\mathbb{P} - e^{f(u,v)} \cdot d\mathbb{Q} = 0$, we can get the optimal mapping function $f^*(u,v) = \log \frac{d\mathbb{P}}{d\mathbb{Q}}$. Rewrite the Equation 34 with $f^*$:

$$\mathbb{E}_\mathbb{P}\left[f^*(u,v)\right] - \mathbb{E}_\mathbb{Q}\left[e^{f^*(u,v)}\right] + 1 = \mathbb{E}_\mathbb{P}\left[\log \frac{d\mathbb{P}}{d\mathbb{Q}}\right] = D_{\text{KL}}(\mathbb{P}\|\mathbb{Q}) \quad (35)$$

LEMMA D.2. *Let $U$, $V$, and $Y$ be three random variables whose sample spaces are $\mathcal{U}, \mathcal{V}$ and $\mathcal{Y}$, $f : (\mathcal{U} \times \mathcal{V} \times \mathcal{Y}) \to \mathbb{R}$ be a mapping function, and $\mathbb{P}$ and $\mathbb{Q}$ be the probability measures on $\mathcal{U} \times \mathcal{V} \times \mathcal{Y}$, we can obtain:*

$$D_{\text{KL}}(\mathbb{P}\|\mathbb{Q}) = \sup_f \mathbb{E}_{(u,v,y) \sim \mathbb{P}}[f(u,v,y)]$$

$$- \mathbb{E}_{(u,v,y) \sim \mathbb{Q}}\left[e^{f(u,v,y)}\right] + 1 \quad (36)$$

*Proof.* The second-order functional derivative of the above function is $-e^{f(u,v,y)} \cdot d\mathbb{Q}$. This negative term means Equation 36 has a supreme value. Through setting the first-order functional derivative as zero $d\mathbb{P} - e^{f(u,v,y)} \cdot d\mathbb{Q} = 0$, we can get the optimal mapping function $f^*(u,v,y) = \log \frac{d\mathbb{P}}{d\mathbb{Q}}$. Rewrite the Equation 36 with $f^*$:

$$\mathbb{E}_\mathbb{P}\left[f^*(u,v,y)\right] - \mathbb{E}_\mathbb{Q}\left[e^{f^*(u,v,y)}\right] + 1 = \mathbb{E}_\mathbb{P}\left[\log \frac{d\mathbb{P}}{d\mathbb{Q}}\right]$$

$$= D_{\text{KL}}(\mathbb{P}\|\mathbb{Q}) \quad (37)$$

## D.2 Results based on Lemma D.1

LEMMA D.3.

$$Weak\text{-}CMI(U; V \mid Y)$$

$$= D_{\text{KL}}\left(P_{U,V}\|\mathbb{E}_{P_Y}\left[P_{U|Y}P_{V|Y}\right]\right)$$

$$= \sup_f \mathbb{E}_{(u,v) \sim P_{U,V}}[f(u,v)] \quad (38)$$

$$- \mathbb{E}_{(u,v) \sim \mathbb{E}_{P_Y}[P_{U|Y}P_{V|Y}]}\left[e^{f(u,v)}\right] + 1$$

*Proof.* Let $\mathbb{P}$ be the joint distribution $P_{U,V}$ and $\mathbb{Q}$ be expectation on the product of marginal distribution $\mathbb{E}_{P_Y}\left[P_{U|Y}P_{V|Y}\right]$ in Lemma D.1.

LEMMA D.4.

$$\sup_f \mathbb{E}_{(u,v_1) \sim \mathbb{P}, (u,v_{2:n}) \sim \mathbb{Q}^{\otimes(n-1)}} \left[\log \frac{e^{f(u,v_1)}}{\frac{1}{n}\sum_{j=1}^n e^{f(u,v_j)}}\right] \quad (39)$$

$$\leq D_{\text{KL}}(\mathbb{P}\|\mathbb{Q})$$

*Proof.* $\forall f$, we can draw:

$$D_{\text{KL}}(\mathbb{P}\|\mathbb{Q}) = \mathbb{E}_{(u,v_{2:n}) \sim \mathbb{Q}^{\otimes(n-1)}}\left[D_{\text{KL}}(\mathbb{P}\|\mathbb{Q})\right]$$

$$\geq \mathbb{E}_{(u,v_{2:n}) \sim \mathbb{Q}^{\otimes(n-1)}}\left[\mathbb{E}_{(u,v_1) \sim \mathbb{P}}\left[\log \frac{e^{f(u,v_1)}}{\frac{1}{n}\sum_{j=1}^n e^{f(u,v_j)}}\right]\right.$$

$$\left. - \mathbb{E}_{(u,v_1) \sim \mathbb{Q}}\left[\frac{e^{f(u,v_1)}}{\frac{1}{n}\sum_{j=1}^n e^{f(u,v_j)}}\right] + 1\right]$$

$$= \mathbb{E}_{(u,v_{2:n}) \sim \mathbb{Q}\otimes(n-1)}\left[\mathbb{E}_{(u,v_1) \sim \mathbb{P}}\left[\log \frac{e^{f(u,v_1)}}{\frac{1}{n}\sum_{j=1}^n e^{f(u,v_j)}}\right]\right.$$

$$\left. - 1 + 1\right] \quad (40)$$

$$= \mathbb{E}_{(u,v_1) \sim \mathbb{P}, (u,v_{2:n}) \sim \mathbb{Q}\otimes(n-1)}\left[\log \frac{e^{f(u,v_1)}}{\frac{1}{n}\sum_{j=1}^n e^{f(u,v_j)}}\right].$$

In detail, the first line always exists because $D_{\text{KL}}(\mathbb{P}\|\mathbb{Q})$ is a constant. The second line comes from Lemma D.1. And because $(u,v_1)$ and $(u,v_{2:n})$ can be interchangeable when they are all sampled from $\mathbb{Q}$, we can obtain the result in the third line. In conclusion, since the inequality works for $\forall f$, we can obtain Lemma D.4

## D.3 Results based on Lemma D.2

LEMMA D.5.

$$CMI(U; V \mid Y) := \mathbb{E}_{P_Y}\left[D_{\text{KL}}\left(P_{U,V|Y}\|P_{U|Y}P_{V|Y}\right)\right]$$

$$= D_{\text{KL}}\left(P_{U,V,Y}\|P_Y P_{U|Y}P_{V|Y}\right)$$

$$= \sup_f \mathbb{E}_{(u,v,y) \sim P_{U,V,Y}}[f(u,v,y)] \quad (41)$$

$$- \mathbb{E}_{(u,v,y) \sim P_Y P_{U|Y}P_{V|Y}}\left[e^{f(u,v,y)}\right] + 1$$

## D.4 Proving Weak-CMI $(U; V | Y) <$ CMI$(U; V | Y)$

PROPOSITION D.6. *Weak-CMI* $(U; V | Y) \leq$ *CMI*$(U; V | Y)$.

*Proof.* According to Lemma D.3,

Weak-CMI$(U; V | Y)$

$$
\begin{aligned}
&= \sup_f \mathbb{E}_{(u,v) \sim P_{U,V}} \left[ f(u, v) \right] \\
&\quad - \mathbb{E}_{(u,v) \sim \mathbb{E}_{P_Y} \left[ P_{U|Y} P_{V|Y} \right]} \left[ e^{f(u,v)} \right] + 1 \\
&= \sup_f \mathbb{E}_{(u,v,y) \sim P_{U,V,Y}} \left[ f(u, v) \right] \\
&\quad - \mathbb{E}_{(u,v,y) \sim P_Y P_{U|Y} P_{U|Y}} \left[ e^{f(u,v)} \right] + 1
\end{aligned}
\tag{42}
$$

When the equality for Weak-CMI holds, we assume the function as $f_1^*(u, v)$. And let $f_2^*(u, v, y) = f_1^*(u, v)$ which means $\forall y \sim P_Y$, $f_2^*(u, v, y)$ will not change. Then, we can get:

Weak-CMI$(U; V | Y)$

$$
\begin{aligned}
&= \mathbb{E}_{(u,v,y) \sim P_{U,V,Y}} \left[ f_1^*(u, v) \right] \\
&\quad - \mathbb{E}_{(u,v,y) \sim P_Y P_{U|Y} P_{V|Y}} \left[ e^{f_1^*(u,v)} \right] + 1 \\
&= \mathbb{E}_{(u,v,y) \sim P_{U,V,Y}} \left[ f_2^*(u, v, y) \right] \\
&\quad - \mathbb{E}_{(u,v,y) \sim P_Y P_{U|Y} P_{U|Y}} \left[ e^{f_2^*(u,v,y)} \right] + 1
\end{aligned}
\tag{43}
$$

Comparing Equation 43 with Lemma D.5, we can conclude Weak-CMI $(U; V | Y) \leq$ CMI$(U; V | Y)$. □

## D.5 Showing the Equation 2 is a lower bound of CMI

PROPOSITION D.7. *We restate the Equation 2 in the main text, and call it as the estimate of CMI (CMIE):*

$$
\begin{aligned}
CMIE &:= \sup_f \mathbb{E}_{y \sim P_Y} \left[ \mathbb{E}_{(u_i,v_i) \sim P_{U,V|y} \otimes n} \left[ \log \frac{e^{f(u_i,v_i)}}{\frac{1}{n} \sum_{j=1}^n e^{f(u_i,v_j)}} \right] \right] \\
&\leq D_{\mathrm{KL}} \left( P_{U,V} \| \mathbb{E}_{P_Y} \left[ P_{U|Y} P_{V|Y} \right] \right) \\
&= \textit{Weak-CMI}\ (U; V | Y) \leq \textit{CMI}(U; V | Y)
\end{aligned}
\tag{44}
$$

*Proof.* By defining $\mathbb{P} = P_{U,V}$ and $\mathbb{Q} = \mathbb{E}_{P_Y} \left[ P_{U|Y} P_{V|Y} \right]$ we can obtain:

$$
\begin{aligned}
&\mathbb{E}_{(u,v_1) \sim \mathbb{P}, (u,v_{2:n}) \sim \mathbb{Q} \otimes (n-1)} \left[ \log \frac{e^{f(u,v_1)}}{\frac{1}{n} \sum_{j=1}^n e^{f(u,v_j)}} \right] = \\
&\mathbb{E}_{y \sim P_Y} \left[ \mathbb{E}_{(u_i,v_i) \sim P_{U,V|y} \otimes n} \left[ \log \frac{e^{f(u_i,v_i)}}{\frac{1}{n} \sum_{j=1}^n e^{f(u_i,v_j)}} \right] \right]
\end{aligned}
\tag{45}
$$

Combined with Lemma D.4, we can deduce:

$$
\begin{aligned}
&\sup_f \mathbb{E}_{y \sim P_Y} \left[ \mathbb{E}_{(u_i,v_i) \sim P_{U,V|y} \otimes n} \left[ \log \frac{e^{f(u_i,v_i)}}{\frac{1}{n} \sum_{j=1}^n e^{f(u_i,v_j)}} \right] \right] \leq \\
&\qquad\qquad D_{\mathrm{KL}} \left( P_{U,V} \| \mathbb{E}_{P_Y} \left[ P_{U|Y} P_{V|Y} \right] \right)
\end{aligned}
\tag{46}
$$

Through Proposition 42 that Weak-CMI $(U; V | Y) \leq$ CMI$(U; V | Y)$, we can draw the conclusion that CMIE is a lower bound of CMI. □

## E ONLINE CLUSTERING

To avoid the trivial solution, we add the constraint that the prototype assignments must be equally partitioned following:

$$
Q = \left\{ Q \in \mathbb{R}_+^{K \times B} \mid Q \mathbb{1}_B = \frac{1}{K} \mathbb{1}_K, Q^\top \mathbb{1}_K = \frac{1}{B} \mathbb{1}_B \right\},
\tag{47}
$$

where the matrix $Q = [q_{u_1}, q_{u_2}, \cdots, q_{u_B}]$ will be optimized that belong to *transportation polytope*, $\mathbb{1}_K$, $\mathbb{1}_B$ denotes the vectors of all ones containing K, B dimension. Then, the objective function of Equation 17 can be reformulated as

$$
\min_{p,q} \mathcal{L}_{\mathrm{clu}} = \min_{Q \in Q} \langle Q, -\log P \rangle_{\mathrm{F}} - \log B,
\tag{48}
$$

where $P = [\frac{1}{B} p_{u_1}, \cdots, \frac{1}{B} p_{u_B}]$ is calculated by Equation 16 and $\langle \cdot \rangle_{\mathrm{F}}$ is the Frobenius dot-product. The loss function in Equation 48 is an *optimal transport problem* that can be efficiently addressed by iterative *Sinkhorn-Knopp algorithm* [12]:

$$
Q^* = \mathrm{Diag}(t) \exp \left( P^\lambda \right) \mathrm{Diag}(r),
\tag{49}
$$

where $t$ and $r$ are renormalization vectors which are calculated by the iterative Sinkhorn-Knopp algorithm, and the hyper-parameter $\lambda$ is employed to balance the convergence speed and the proximity to the original transport problem.

## F ALGORITHM

The augmentation process consists of two main steps. First, the augmentation function (which includes feature masking and edge dropping) is applied to the original graph, resulting in two augmented views. Second, a learnable perturbation is added to one of the views' input feature spaces. These two augmented views are then fed into a shared encoder to obtain node representations. The final step involves applying the contrastive loss to these representations. To optimize the perturbation, we maximize the contrastive loss in the inner loop while accumulating gradients to optimize the encoder parameters in the outer loop. The complete pipeline is detailed in Algorithm 1.

## G EXPERIMENTAL DETAILS

### G.1 Baselines

We consider empirical risk minimization (ERM), one OOD algorithm IRM and one graph-specific OOD algorithm EERM as supervised baselines. And we include eleven self-supervised methods as unsupervised baselines:

- **Invariant Risk Minimization (IRM [4])** is an algorithm that seeks to learn data representations that are robust and generalize well across different environments by penalizing feature distributions that have different optimal linear classifiers for each environment
- **EERM [64]** generates multiple graphs by environment generators and minimizes the mean and variance of risks from multiple environments to capture invariant features.
- **Graph Autoencoder (GAE [32])** is an encoder-decoder structure model. Given node attributes and structures, the encoder will compress node attributes into low-dimension latent space, and the decoder (dot-product) hopes to reconstruct existing links with compact node features.

**Algorithm 1** Algorithm for a shift-robust framework for graph contrastive learning.

---

**Input**: Augmentation pool $\mathcal{T}$, ascent steps $M$, ascent step size $\epsilon$, encoder $g_\theta$, projector $p_\omega$, and training graph $G = (A, X)$

1: **while** not converge **do**
2:     $\tau_\alpha, \tau_\beta \sim \mathcal{T}$
3:     $G_\alpha, G_\beta = \tau_\alpha(G), \tau_\beta(G)$
4:     $C = \arg\min_C \mathcal{L}_{\text{clu}}\left(g(G_\alpha), g(G_\beta), C\right)$
5:     $\delta_0 \leftarrow U(-\epsilon, \epsilon)$
6:     $\hbar_0 \leftarrow 0$
7:     **for** t = 1 … $M$ **do**
8:       $Z_\alpha = p_\omega \circ g_\theta \left(X_\alpha + \delta_{t-1}, A_\alpha\right)$
9:       $Z_\beta = p_\omega \circ g_\theta(X_\beta, A_\beta)$
10:      $\hbar_t \leftarrow \hbar_{t-1} + \frac{1}{M} \cdot \nabla_{\theta, \omega} \mathcal{L}_{\text{rob}}\left(Z_\alpha, Z_\beta, C\right)$
11:      $\hbar_\delta \leftarrow \nabla_\delta \mathcal{L}_{\text{rob}}\left(Z_\alpha, Z_\beta, C\right)$
12:      $\delta_t \leftarrow \delta_{t-1} + \epsilon \cdot \hbar_\delta / \|\hbar_\delta\|_F$
13:     **end for**
14:     $\theta \leftarrow \theta - \eta \cdot \hbar_{M, \theta}$
15:     $\omega \leftarrow \omega - \eta \cdot \hbar_{M, \omega}$
16: **end while**

---

- Variational Graph Autoencoder (VGAE [32]) is similar to GAE but the node features are re-sampled from a normal distribution through a re-parameterization trick.
- GraphMAE [25] is a masked autoencoder. Different to GAE and VGAE, it will mask partial input node attributes firstly and then the encoder will compress the masked graph into latent space, finally a decoder aims to reconstruct the masked attributes.
- Deep Graph Infomax (DGI [62]) is a node-graph contrastive method which contrasts the node representations and graph representation. First, it will apply the corrupt function to obtain a negative graph and two graphs will be fed into a shared GNN model to generate node embeddings. And a readout function will be applied on the original node embeddings to obtain graph-level representation. Corrupted embeddings and readout graph representation are considered as positive pairs, original node representations and readout graph representation are considered as positive pairs.
- MVGRL [22] is similar to DGI but utilizes the information of multi-views. Firstly, it will use edge diffusion function to generate an augmented graph. And asymmetric encoders will be applied on the original graph and diffusion graph to acquire node embeddings. Next, a readout function is employed to derive graph-level representations. Original node representations and augmented graph-level representation are regarded positive pairs. Additionally, the augmented node representations and original graph-level representation are also considered as positive pairs. The negative pairs are constructed following [62].
- RoSA [77] is a robust self-aligned graph contrastive framework which does not require the explicit alignment of nodes

in the positive pairs so that allows more flexible graph augmentation. It proposes the graph earth move distance (g-EMD) to calculate the distance between unaligned views to achieve self-alignment. Furthermore, it will use adversarial training to realize robust alignment.
- GRACE [78] is node-node graph contrastive learning method. It designs two augmentation functions (*i.e.*, removing edges and masking node features) to generate two augmented views. Then a shared graph model will be applied on augmented views to generate node embedding matrices. The node representations augmented from the same original node are regarded as positive pairs, otherwise are negative pairs. Lastly, pairwise loss (*e.g.*, InfoNCE [47]) will be applied on these node matrices.
- BGRL [57] is similar to GRACE but without negative samples which is motivated by BYOL [18].
- G-BT [7] is a self-supervised framework that introduces Barlow Twins [71] into graph domain. This method also gets rid of negative samples.
- COSTA [73] proposes feature augmentation to decrease the bias introduced by graph augmentation.
- SwAV [9] is an unsupervised online clustering method which incorporates prototypes for clustering and employs swapped prediction for model training. It is originally designed for the computer vision domain, we adopt it into graph domain.

## G.2 Datasets

For GOOD-Cora, GOOD-Twitch, GOOD-CBAS and GOOD-WebKB datasets, they are all adopted from GOOD[19] which is a comprehensive Graph OOD benchmark. These datasets contain both concept shift and covariate shift splits, for more details of splitting, please refer to Appendix A in [19].

GOOD-Cora is a citation network that is derived from the full Cora dataset [8]. In the network, each node represents a scientific publication, and edges between nodes denote citation links. The task is to predict publication types (70-classification) of testing nodes. The data splits are generated based on two domain selections (*i.e.*, word, degree). The word diversity selection is based on the count of selected words within a publication and is independent of the publication's label. On the other hand, the node degree selection ensures that the popularity of a paper does not influence its assigned class.

GOOD-Twitch is a gamer network dataset. In this dataset, each node represents a gamer, and the node features correspond to the games played by each gamer. The edges between nodes represent friendship connections among gamers. The binary classification task associated with this dataset involves predicting whether a user streams mature content or not. The data splits for GOOD-Twitch are based on the user language, ensuring that the prediction target is not biased by the specific language used by a user.

GOOD-CBAS is a synthetic dataset that is modified from the BA-Shapes dataset [69]. It involves a graph where 80 house-like

---

[5]This dataset is adopted from [68]. [64] constructs ten graphs with different environment id's for each graph.
[6]The original is available on https://www.kaggle.com/ellipticco/elliptic-data-set

**Table 4: The descriptions of datasets. "Domain-Level" means splitting by graphs, "Time-Aware" denotes splitting according to chronological order."Word" and "Degree" represent splitting according to word diversity and node degree respectively. "Language" means splitting by user language, implying that the prediction target should not be biased by the language a user uses. "University" denotes splitting according to the domain university, suggesting that classified webpages are based on word contents and link connections instead of university features. "Color" means splitting by node color differences in covariate splits and color-label correlations in concept splits.**

| Datasets | Network Type | #Nodes | #Edges | #Attributes | #Classes | Train/Val/Test Split | Metric |
|---|---|---|---|---|---|---|---|
| Amazon-Photo[7] | Co-purchasing network | 7,650 | 119,081 | 755 | 10 | Domain-Level | Accuracy |
| Elliptic[8] | Bitcoin transactions | 203,769 | 234,355 | 165 | 2 | Time-Aware | F1-Score |
| GOOD-Cora | Scientific publications | 19,793 | 126,842 | 8,710 | 70 | Word/Degree | Accuracy |
| GOOD-Twitch | Gamer network | 34,120 | 892,346 | 128 | 2 | Language | ROC-AUC |
| GOOD-CBAS | A BA-house graph | 700 | 3,962 | 4 | 4 | Color | Accuracy |
| GOOD-WebKB | Webpage network | 617 | 1,138 | 1,703 | 5 | University | Accuracy |

motifs are attached to a base graph following the Barabási–Albert model, resulting in a graph with 300 nodes. The task associated with this dataset is to predict the role of each node within the graph. The roles can be classified into four classes, which include identifying whether a node belongs to the top, middle, or bottom of a house-like motif, or if it belongs to the base graph itself. In contrast to using constant node features, the GOOD-CBAS dataset introduces colored features. This modification poses challenges for out-of-distribution (OOD) algorithms, as they need to handle differences in node colors within covariate splits and consider the correlations between node color and node labels within concept splits.

GOODWebKB is a network dataset that focuses on university webpages. Each node in the network represents a webpage, and the node features are derived from the words that appear on the webpage. The edges between nodes represent hyperlinks between webpages. The task associated with this dataset is a 5-class prediction task, where the goal is to predict the class of each webpage. The data splits for GOOD-WebKB are based on the domain of the university, ensuring that the classification of webpages is based on their word contents and link connections rather than any specific university features.

Amazon-Photo is a co-purchasing network that is widely used for evaluating the design of GNN models. In this network, each node corresponds to a specific product, and the presence of an edge between two products indicates that they are frequently purchased together by customers. In the original dataset, it is observed that the node features exhibit a significant correlation with the corresponding node labels. In order to evaluate the model's ability to generalize to out-of-distribution scenarios, it is necessary to introduce distribution shifts into the training and testing data. To achieve this, we adopt the strategies employed in the EERM [64]. Specifically, we leverage the available node features $X_1$ to create node labels $Y$ and spurious environment-sensitive features $X_2$. To elaborate, a randomly initialized GNN takes $X_1$ and the adjacency matrix as inputs and employs an argmax operation in the output layer to obtain one-hot vectors as node labels. Additionally, we employ another randomly initialized GNN that takes the concatenation of $Y$ and an environment id as input to generate spurious node features $X_2$. By combining these two sets of features, we obtain the input node features, $X = [X_1, X_2]$, which are used for both training

and evaluation. This process is repeated to create ten graphs with distinct environment id's. Such a shift between different graphs can be considered as a concept shift [55]. Finally, one graph is allocated for training, another for validation, and the remaining graphs are used for evaluating the OOD generalization of the trained model.

Elliptic is a financial network that records the payment flows among transactions as time goes by. It consists of 49 graph snapshots which are collected at different times. Each graph snapshot is a network of Bitcoin transactions where each node represents one transaction and each edge denotes a payment flow. Partial nodes (approximately 20%) are labeled as licit or illicit transactions and we hope to identify illicit transactions in the future. For data preprocessing, we adopt the same strategies in EERM [64]: removing extremely imbalanced snapshots and using the 7th-11th/12th-17th/17th-49th snapshots for training/validation/testing data. And 33 testing graph snapshots will be split into 9 test sets according to chronological order. In Figure 6, we depict the label rate and positive label rate for training/validation/testing sets. It is evident that the varying positive label rates across different data sets are apparent. Indeed, the model needs to deal with the label distribution shifts from training to testing data.

## G.3 Experimental Setup

**Transductive setting.** We use the same model configuration (*e.g.*, same encoder) across different datasets for a fair comparison following [19]. We use grid search to find other hyper-parameters (*e.g.*, learning rate, epochs) for different methods. For all experiments, we select the best checkpoints for ID and OOD tests according to results on ID and OOD validation sets following [19], respectively. Experimental details and hyper-parameter selections are provided in Appendix G.4. For evaluating unsupervised methods, a linear classifier will be built on the frozen trained encoder after finishing pre-training. The reported results are the mean performance with standard deviation after 10 runs following [19].

**Inductive setting.** For GOOD-WebKB and GOOD-CBAS datasets, we use the same model configuration in Section 4.2.1. For Amazon-Photo dataset [68] and Elliptic [48] dataset, they consist of many snapshots (training data and testing data use different snapshots) which are naturally inductive. For Amazon-Photo dataset, we use

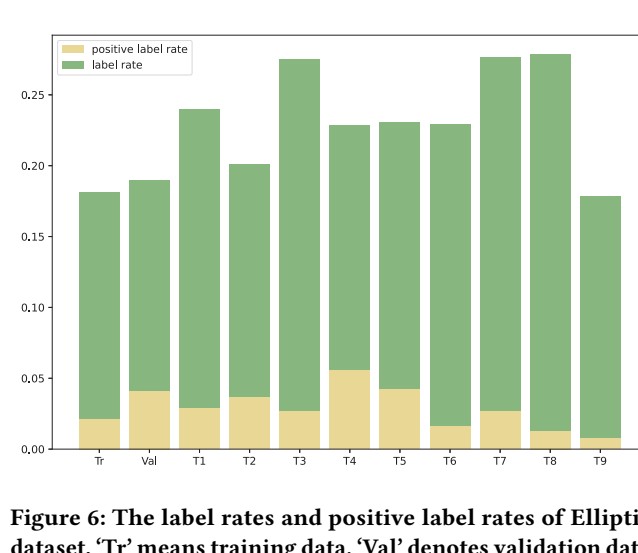

**Figure 6: The label rates and positive label rates of Elliptic dataset. 'Tr' means training data, 'Val' denotes validation data and others represent testing data. In the different splits, the label distributions are disparate.**

2-layer GCN [33] as the encoder and for elliptic dataset, we use 5-layer GraphSAGE [21] as encoder following [64].

## G.4 Hyper-parameters

For GOOD datasets, we adopt GraphSAINT [72] as subsampling technique, while utilizing a 3-layer GCN [33] with 300 hidden units as the backbone following [19]. For the supervised baselines (*i.e.*, ERM, IRM, EERM), we use the identical hyper-parameters specified in [19]. For other unsupervised baselines, we conduct a grid search to find the best performance. Specifically, max training epoch ranges in {50, 100, 200, 500, 600} and learning rate ranges in {1e-1,1e-2,1e-3,1e-4,1e-5}, augmentation ratio range in [0.1, 0.6]. Regarding GraphMAE, the masking ratio ranges in {0.25,0.5,0.75}, and we use a one-layer GCN as the decoder. For MARIO, the specific hyper-parameters are listed in Table 5. For the adversarial augmentation, we set ascent steps $M$ as 3 and the ascent step size $\epsilon$ as 1e-3.

For Amazon-Photo, we utilize 2-layer GCN with 128 hidden units as encoder, and we set $\tau, p_{f,1}, p_{f,2}, p_{e,1}, p_{e,2}, \gamma, |C|$ as 0.2, 0.2, 0.3, 0.2, 0.3, 0.1, 100 respectively and learning rate as 1e-4. Other hyper-parameters remain the same in EERM [64]. Regarding Elliptic datasets, we employ 5-layer GraphSAGE [21] with 32 hidden units as encoder following EERM [64]. And we set $\tau, p_{f,1}, p_{f,2}, p_{e,1}, p_{e,2}, \gamma, |C|$ as 0.8, 0.2, 0.3, 0.2, 0.3, 0.5, 120 respectively. The remaining hyper-parameters are consistent with EERM [64].

## G.5 Evaluation metrics

In order to evaluate the pre-trained models, we adopt the linear evaluation protocol which is commonly used in self-supervised methods [10, 18, 23]. That is, we will train a linear classifier (*i.e.*, one-layer MLP) on top of (frozen) representations learned by self-supervised methods. The training epochs (Epochs for LC in Table 5) and learning rate (LR for LC in Table 5) of the linear classifier

are obtained by grid search. The reported results are the mean performance with standard deviation after 10 runs following [19].

## G.6 Computer infrastructures specifications

For hardware, all experiments are conducted on a computer server with eight GeForce RTX 3090 GPUs with 24GB memory and 64 AMD EPYC 7302 CPUs. And our models are implemented by Pytorch Geometric 2.0.4 [16] and Pytorch 1.11.0 [49]. All datasets used in our work are available on https://github.com/divelab/GOOD and https://github.com/qitianwu/GraphOOD-EERM.

## H ADDITIONAL EXPERIMENTS

Due to space constraints in the main content, we have included additional experiments in this section to provide a comprehensive evaluation. Firstly, we perform graph classification in Section H.1. Then, we present additional experiments on the GOOD-CBAS and GOOD-WebKB datasets under the inductive setting. Furthermore, we demonstrate the model-agnostic nature of the recipe by integrating it with various graph neural networks (GNNs), including GCN, GraphSAGE, and GAT, in Section H.3. Additionally, we evaluate our recipe on various self-supervised methods to illustrate that MARIO is a plug-in that can be integrated with many current GCL methods in Section H.4. Finally, we visualize the experimental results, including metric score curves and feature embeddings, in Section H.5.

## H.1 Graph classification

*H.1.1 Datasets.* The PROTEINS dataset comprises protein data. During training, we use graphs ranging from 4 to 25 nodes, while during testing, we evaluate on graphs spanning from 6 to 620 nodes [34]. The D&D dataset is also protein-based and involves two distinct splitting methods, namely D&D$_{200}$ and D&D$_{300}$ [34]. For the D&D$_{200}$ split, training is conducted on graphs containing 30 to 200 nodes, and testing is performed on graphs consisting of 201 to 5,748 nodes. As for the D&D$_{300}$ split, training is carried out on 500 graphs ranging from 30 to 300 nodes, while testing is conducted on other graphs comprising 30 to 5,748 nodes.

*H.1.2 Experimental setups.* For all graph classification datasets, we utilize 2-layer GCN containing 64 hidden units as graph encoder, and we choose global max pooling as the readout function. For other hyper-parameters, we set lr, $\tau, p_{f,1}, p_{f,2}, p_{e,1}, p_{e,2}, \gamma, |C|$ as 0.01, 0.5, 0.2, 0.3, 0.2, 0.3, 0.5, 40 respectively for self-supervised methods. For ERM, the learning rate ranges in {1e-2, 1e-3, 5e-3, 1e-4} and the number of training epochs is selected in {50, 100, 200}. For evaluating pre-trained models, we use an off-the-shelf $\ell_2$-regularized LogisticRegression classifier from Scikit-Learn [50] using the 'liblinear' solver with a small hyperparameter search over the regularization strength to be between $\{2^{-10}, 2^{-9}, \ldots 2^9, 2^{10}\}$.

*H.1.3 Baselines.* GraphCL [70] investigated the impact of different graph augmentations (*i.e.*, node dropping, edge perturbation, attribute masking and subgraph sampling) on graph classification datasets. The framework is similar to SimCLR [10] but specified for graph domain.

*H.1.4 Results.* From Table 6, we can find GraphCL [70] can achieve comparable performance with ERM even without labels. And our

**Table 5: Hyperparameters specifications for MARIO**

| | GOOD-Cora | | | | GOOD-CBAS | | GOOD-Twitch | | GOOD-WebKB | |
| --- | --- | --- | --- | --- | --- | --- | --- | --- | --- | --- |
| | word | | degree | | color | | language | | university | |
| | concept | covariate | concept | covariate | concept | covariate | concept | covariate | concept | covariate |
| Model | GCN | GCN | GCN | GCN | GCN | GCN | GCN | GCN | GCN | GCN |
| # Layers | 3 | 3 | 3 | 3 | 3 | 3 | 3 | 3 | 3 | 3 |
| # Hidden size | 300 | 300 | 300 | 300 | 300 | 300 | 300 | 300 | 300 | 300 |
| Epochs | 100 | 150 | 100 | 100 | 200 | 500 | 600 | 200 | 100 | 500 |
| Learning rate | 1e-3 | 1e-3 | 1e-3 | 1e-3 | 1e-1 | 1e-2 | 1e-1 | 1e-1 | 1e-2 | 1e-2 |
| $\tau$ | 0.2 | 0.2 | 0.2 | 0.2 | 0.2 | 0.2 | 0.2 | 0.2 | 0.5 | 0.5 |
| $p_{f,1}$ | 0.3 | 0.3 | 0.0 | 0.3 | 0.2 | 0.2 | 0.2 | 0.2 | 0.5 | 0.2 |
| $p_{f,2}$ | 0.4 | 0.3 | 0.0 | 0.3 | 0.3 | 0.3 | 0.3 | 0.3 | 0.5 | 0.3 |
| $p_{e,1}$ | 0.4 | 0.4 | 0.6 | 0.6 | 0.2 | 0.2 | 0.2 | 0.2 | 0.5 | 0.2 |
| $p_{e,2}$ | 0.5 | 0.4 | 0.6 | 0.6 | 0.3 | 0.3 | 0.3 | 0.3 | 0.5 | 0.3 |
| $|C|$ | 100 | 100 | 150 | 150 | 100 | 100 | 100 | 100 | 100 | 100 |
| $\gamma$ | 0.2 | 0.5 | 0.8 | 0.8 | 0.1 | 0.1 | 0.2 | 0.2 | 0.1 | 0.2 |
| Pro. LR | 1e-5 | 1e-5 | 1e-5 | 1e-5 | 1e-4 | 1e-7 | 1e-5 | 1e-3 | 1e-3 | 1e-3 |
| Epochs for LC | 500 | 200 | 100 | 100 | 100 | 2000 | 100 | 100 | 50 | 50 |
| LR for LC | 1e-4 | 1e-3 | 1e-3 | 1e-3 | 1e-2 | 1e-3 | 1e-1 | 1e-1 | 1e-1 | 1e-1 |

recipe can boost the OOD generalization ability of unsupervised methods and even surpasses ERM which means our recipe is also effective for graph classification task.

**Table 6: Results of different methods on OOD graph classification tasks. We report the mean of Accuracy with standard deviation after 10 runs.**

| | PROTEIN$_{25}$ | D&D$_{200}$ | D&D$_{300}$ |
| --- | --- | --- | --- |
| #Train/Test Graphs | 500/613 | 462/716 | 400/678 |
| #Nodes Train | 4-25 | 30-200 | 30-300 |
| #Nodes Test | 6-620 | 201-5748 | 30-5748 |
| ERM | 77.24±0.95 | 44.25±5.16 | 67.91±1.60 |
| GraphCL | 76.92±0.91 | 48.12±6.43 | 67.82±1.29 |
| GraphCL(+MARIO) | **78.08±0.97** | **51.62±5.47** | **69.13±1.23** |

## H.2 Inductive Setting on More datasets

In the main content, we conduct experiments on Amazon and Elliptic datasets under the inductive setting. In this subsection, we will supplement experiments of more datasets (*i.e.*, GOOD-WebKB and GOOD-CBAS) under inductive settings.
**Baselines:** For GOOD-WebKB and GOOD-CBAS, we adopt ERM, IRM, GraphMAE, and GRACE as baselines.
**Experimental Setup:** We use the same model configuration in Section 4.2.1. The reported results are the mean performance after 10 runs following [19].
**Analysis:** Based on the observations from Figure 7 and Figure 8 MARIO demonstrates the best performances on both ID and OOD test sets for GOOD-WebKB and GOOD-CBAS datasets, under both concept shift and covariate shift. Notably, MARIO outperforms other methods by more than 3% and 10% absolute improvement

on GOOD-WebKB and GOOD-CBAS, respectively, under covariate shift. We can draw similar conclusions as discussed in Section 4.2.1. Even under the inductive setting, our method continues to demonstrate excellent OOD generalization capabilities and achieves comparable or even improved in-distribution test performance. These statistical results further validate the effectiveness of our method in handling distribution shifts and enhancing generalization performance.

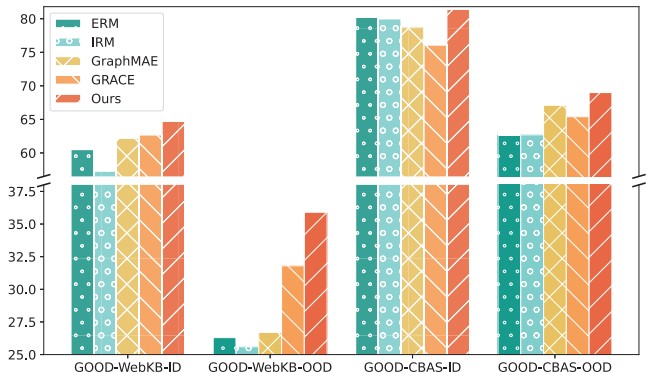

**Figure 7: Results on GOOD-WebKB and GOOD-CBAS datasets with concept shift under the inductive setting. 'GOOD-WebKB-ID' means in-distribution test performance and 'GOOD-WebKB-OOD' means out-of-distribution test performance. So are 'GOOD-CBAS-ID' and 'GOOD-CBAS-OOD'. We report the mean accuracy across 10 runs.**

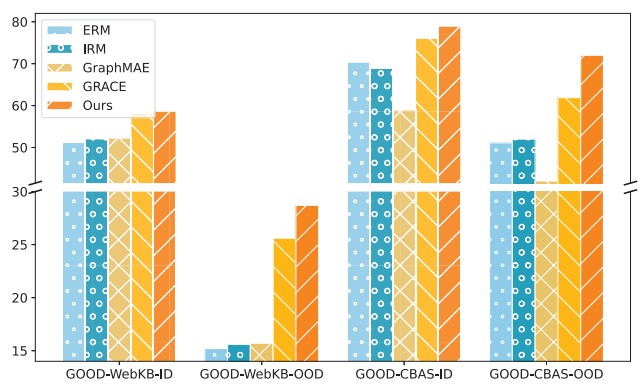

Figure 8: Results on GOOD-WebKB and GOOD-CBAS datasets with covariate shift under the inductive setting.

Table 7: Results of different learning approaches with different encoding models (*i.e.,* GCN, GraphSAGE, GAT).

| Model | Method | GOOD-CBAS color | | GOOD-WebKB university | |
|---|---|---|---|---|---|
| | | ID | OOD | ID | OOD |
| GCN | ERM | 89.79±1.39 | 83.43±1.19 | 62.67±1.53 | 26.33±1.09 |
| | GRACE | 92.00±1.39 | 88.64±0.67 | 64.00±3.43 | 34.86±3.43 |
| | MARIO | 94.36±1.21 | 91.28±1.10 | 65.67±2.81 | 37.15±2.37 |
| SAGE | ERM | 95.07±1.51 | 75.14±1.19 | 73.67±2.08 | 46.33±3.42 |
| | GRACE | 95.29±1.11 | 74.43±2.36 | 70.50±5.06 | 49.54±3.83 |
| | MARIO | 96.00±1.07 | 76.29±3.01 | 71.00±3.82 | 51.74±4.63 |
| GAT | ERM | 78.64±3.63 | 72.93±2.64 | 61.33±3.71 | 28.99±2.63 |
| | GRACE | 84.57±1.79 | 78.36±1.60 | 59.50±2.36 | 35.78±3.26 |
| | MARIO | 84.93±1.95 | 80.43±1.89 | 62.17±4.78 | 38.17±3.10 |

## H.3 Integrated with Other Models

In the subsection, we demonstrate the model-agnostic nature of the recipe by integrating it with various graph neural network (GNN) models, including GCN, GraphSAGE, and GAT.

From Table 7, it can be observed that regardless of the specific GNN model used as the encoder, our method consistently achieves the best performance on the OOD test set. This indicates the effectiveness and robustness of our method across different GNN models. By achieving superior performance across different GNN models, MARIO demonstrates its versatility and ability to improve the OOD generalization of various graph neural models. This highlights the broad applicability and effectiveness of our recipe in enhancing the performance of different GNN encoders.

Furthermore, we integrate our recipe with other GCL methods in Appendix H.4. The results demonstrate our recipe can boost the OOD generalization ability of various GCL methods which means our recipe can serve as a plug-in for many current classical GCL methods.

## H.4 Integrated with Other Methods

Our recipe is not only model-agnostic but also an add-on training scheme that can be adopted on most graph contrastive learning

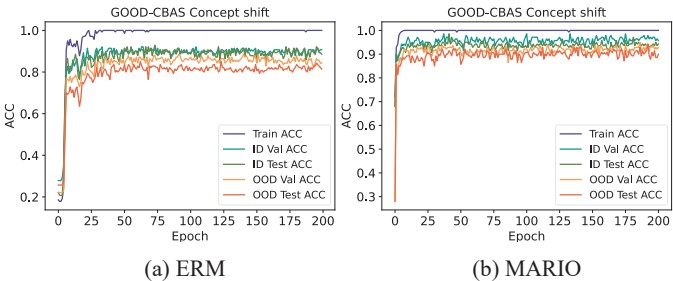

Figure 9: Metric score curves for ERM and MARIO on GOOD-CBAS.

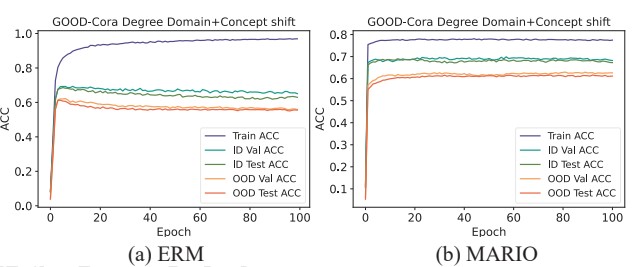

Figure 10: Metric score curves for ERM and MARIO on GOOD-Cora word domain with concept shift.

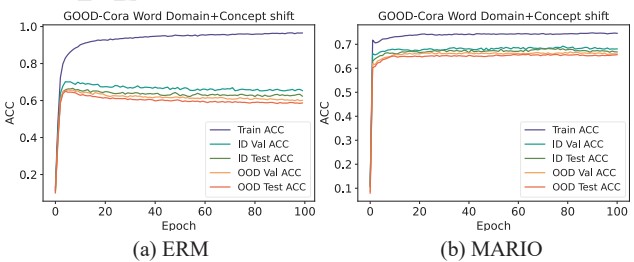

Figure 11: Metric score curves for ERM and MARIO on GOOD-Cora degree domain with concept shift.

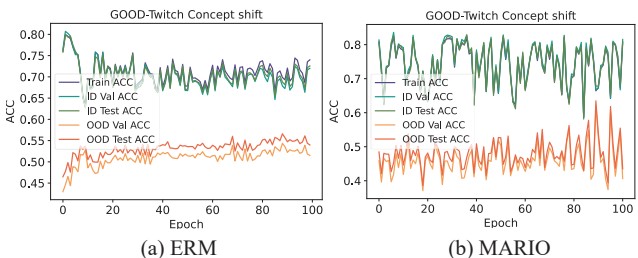

Figure 12: Metric score curves for ERM and MARIO on GOOD-Twitch language domain with concept shift.

(GCL) methods. In Table 8, we use our recipe to guide various GCL methods (GRACE, COSTA). MARIO can further boost these methods on both ID and OOD test performance.

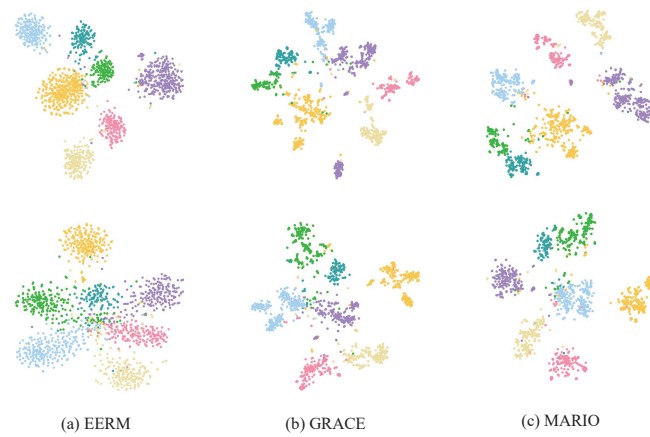

(a) EERM      (b) GRACE      (c) MARIO

**Figure 14: t-SNE visualization of node embeddings on GOOD-Cora dataset[9], (a) depicts node embeddings from trained EERM, (b) shows embeddings from trained GRACE model, (c) is the result of trained MARIO. The margins of each cluster learned from MARIO are much wider than others.**

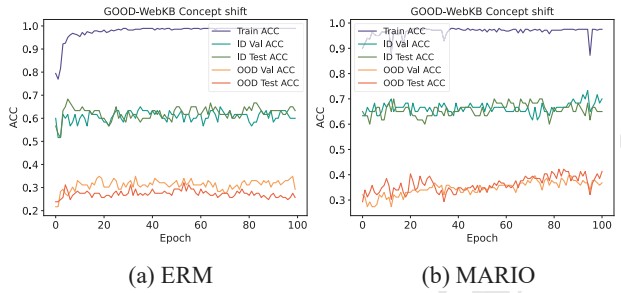

(a) ERM      (b) MARIO

**Figure 13: Metric score curves for ERM and MARIO on GOOD-WebKB university domain with concept shift.**

**Table 8: Results of various methods integrated with MARIO. We report the mean and standard deviation of Accuracy after 10 runs.**

| concept shift | GOOD-WebKB university | | GOOD-CBAS color | |
|---|---|---|---|---|
| | ID | OOD | ID | OOD |
| GRACE | 64.00±3.43 | 34.86±3.43 | 92.00±1.39 | 88.64±0.67 |
| GRACE (+MARIO) | 65.67±2.81 | 37.15±2.37 | 94.36±1.21 | 91.28±1.10 |
| COSTA | 61.66±2.58 | 32.39±2.13 | 93.50±2.62 | 89.29±3.11 |
| COSTA(+MARIO) | 62.33±2.60 | 35.32±3.46 | 98.00±1.31 | 94.36±1.51 |

## H.5 Visualization

*H.5.1 Metric Score Curves.* We present metric score curves for ERM and MARIO, including training, ID validation, ID testing, OOD validation, and OOD testing accuracy, in Figure 9,10,11,12,13. Notably, MARIO demonstrates superior convergence with approximately 10% absolute improvement on the OOD test set compared to ERM on GOOD-CBAS dataset. Furthermore, MARIO effectively narrows the performance gap between in-distribution and out-of-distribution performance, showcasing its efficacy in enhancing OOD generalization for graph data.

*H.5.2 Feature Visualization.* We visualize both in-distribution (ID) and out-of-distribution (OOD) node embeddings separately in Figure 14. The first row shows the ID node embeddings, while the second row displays the OOD node embeddings. Notably, MARIO exhibits superior clustering performance compared to other methods for both ID and OOD node sets. The clusters are more distinct, with larger margins, underscoring the effectiveness of our method in handling OOD data.

Received 20 February 2007; revised 12 March 2009; accepted 5 June 2009

---

[9]In order to better visualization, we select seven informative classes. And all models are trained under concept shift in degree domain.

