# OpenReview forum: "MARIO: Model Agnostic Recipe for Improving OOD Generalization of Graph Contrastive Learning"
_ACM.org/TheWebConf/2024/Conference — TheWebConf24_

### Official Review · Reviewer_Bco1 · 2023-11-13

**Novelty:** 4
**Technical Quality:** 5

**Review:**

This paper describes a new recipe of graph contrastive learning to improve model performance of OOD generalization in unsupervised scenario. The recipe includes two common principles: invariant representation learning and information bottleneck. Empirical results on multiple datasets show the superiority of the proposed framework.

Strengths:
1)This paper is well motivated. Investigating the generalization ability of unsupervised graph learning is an important and interesting direction.
2)This paper includes extensive experiments on graph tasks to verify the effectiveness of the proposed method.
3)This paper gives a detailed background of used techniques to introduce the proposed method.

Weaknesses:
1)The paper is not well written and the organization is unclear. Some used concepts and concrete equations are not well explained, such as the concrete definition of environment labels. Some proofs used in existing work are not the contributions of this paper, so it is confusing to append them in Appendix.
2)The model includes many incremental improvements so that it is a very complicated model, but the runtime complexity of this paper is not given.
3)The reproducible experiments are not provided.

**Questions:**

See above weaknesses.

**Reviewer Confidence:**

4: The reviewer is certain that the evaluation is correct and very familiar with the relevant literature

**Scope:**

4: The work is relevant to the Web and to the track, and is of broad interest to the community

---

### Official Review · Reviewer_96AN · 2023-11-19

**Novelty:** 6
**Technical Quality:** 6

**Review:**

The authors propose a model-agnostic recipe to address the challenges of distribution shifts in graph contrastive learning motivated by invariant learning and information bottleneck principles. The authors conduct sufficient theoretical proofs and experiments. The experimental results also illustrate the validity of the model proposed by the authors.

**Questions:**

1. Do not add watermarks to the draft paper.

2. Since the authors propose many loss functions, one model diagram or a detailed algorithm is necessary to be included in the paper. The current algorithm 1 in the appendix does not clearly show the process of the algorithm.

3. Why not put Citation 40 (Sihang Li, Xiang Wang, An Zhang, Yingxin Wu, Xiangnan He, and Tat-Seng Chua. 2022. Let invariant rationale discovery inspire graph In International Conference on Machine Learning. PMLR, 13052-13065) as a baseline?

**Reviewer Confidence:**

3: The reviewer is confident but not certain that the evaluation is correct

**Scope:**

4: The work is relevant to the Web and to the track, and is of broad interest to the community

---

### Official Review · Reviewer_oqvZ · 2023-11-23

**Novelty:** 6
**Technical Quality:** 6

**Review:**

Contributions:

The authors observe that graph contrastive learning (GCL) methods differ in their robustness to OOD tests. To address this, the authors analyze the limitations of these methods and propose a contrastive learning framework called MARIO to improve robustness against distributional shifts. MARIO comprises adversarial augmentations (to push for invariant representations) and improved representation contrasting. The authors evaluate MARIO's performance on OOD and in-distribution test sets.

My recommendation is based on S1, S2, W1, W2, W3. I am happy to raise my scores based on the authors' responses to my questions and clarification/justification of W1, W2, W3.

Quality:

Pros:
- (S2) The authors' theoretical analysis of augmentations and representation contrasting is interesting and motivates the adversarial augmentation and CMI minimization components of MARIO well.

- (S3) Experiments: The authors evaluate MARIO in transductive and inductive settings and with respect to various relevant supervised, IRM, and graph OOD baselines. In the transductive setting, MARIO mostly outperforms the baselines both in-distribution and OOD for the datasets, but MARIO's accuracy is not significantly higher. MARIO consistently outperforms baselines in the inductive setting.

Cons:
- (W3) The authors should provide further justification for why online clustering labels serve as good pseudo-labels.

- What is the added time complexity of online clustering?

- Experiments: The authors should comment on the validity of using graph OOD benchmarks to evaluate their node OOD method.

Clarity:

Pros:
- The writing is extremely clear and well-organized, with vivid examples and solid interpretations of theoretical results.

Originality:

Pros:
- To the best of the authors' knowledge, they are the first to study the OOD generalization of graph contrastive learning for node-level tasks.

- The authors thoroughly compare/contrast MARIO to related methods.

Cons:
- (W1) The authors adopt FLAG (an existing graph augmentation mechanism) and the common practice of using prototypes in GCL (e.g., [1, 2]).

Significance:

Pros:
- (S1) The authors focus on node-level tasks, which are more challenging due to the interconnected nature of nodes and the diverse types of possible distribution shifts.

Cons:
- (W2) The authors should comment on the limitations of their method.

[1] Zhang, Shichang, et al. "Motif-driven contrastive learning of graph representations." arXiv preprint arXiv:2012.12533 (2020).

[2] Li, Bolian, Baoyu Jing, and Hanghang Tong. "Graph communal contrastive learning." Proceedings of the ACM Web Conference 2022. 2022.

EDIT: I have read the authors' rebuttal.

**Questions:**

Please see Review (above).

**Reviewer Confidence:**

2: The reviewer is willing to defend the evaluation, but it is likely that the reviewer did not understand parts of the paper

**Scope:**

3: The work is somewhat relevant to the Web and to the track, and is of narrow interest to a sub-community

---

### Official Review · Reviewer_HF5w · 2023-11-23

**Novelty:** 6
**Technical Quality:** 6

**Review:**

The Paper proposes a novel model-agnostic recipe for OOD generalization problem of graph contrastive learning methods, which mainly works on the view generation and representation contrasting component of GCL.

Strength

1)The authors investigate the OOD generalization problem of graph contrastive learning specifically for node-level tasks for the first time.

2)Formally formulated the problem of graph contrastive learning for OOD generalization.

3)The experiments are extensive with various baselines to certify the effectiveness of the approach.

4)The paper is well presented and structured.

Weakness

1)It is better to provide a figure of framework or overview for MARIO, which is clearer and more friendly for readers.

2)The sensitivity analysis does not cover all hyperparameters.

**Questions:**

See the weakness.

**Ethics Review Description:**

No issues

**Reviewer Confidence:**

3: The reviewer is confident but not certain that the evaluation is correct

**Scope:**

4: The work is relevant to the Web and to the track, and is of broad interest to the community

---

### Official Review · Reviewer_S5XT · 2023-11-29

**Novelty:** 5
**Technical Quality:** 6

**Review:**

Summary

The paper focuses on graph-structured data applications and the challenges in out-of-distribution (OOD) generalization, particularly with unlabeled graph data. It identifies two primary challenges: the non-Euclidean nature of graphs causing complex distributional shifts and the heavy reliance on label information in existing OOD generalization methods. So the paper introduces MARIO (Model-Agnostic Recipe for Improving OOD generalization of GCL methods), which operates on two key aspects of GCL: view generation and representation contrasting. MARIO integrates the Invariance principle, using adversarial graph augmentation, and the Information Bottleneck principle, aiming to develop robust GCL methods against distributional shifts. Extensive experiments demonstrate that MARIO effectively enhances the OOD generalization capabilities of GCL methods.

Pros

(1) Relevance of Study: The paper tackles important tasks in the domain of unsupervised learning on graph data, focusing on improving OOD generalization.

(2) New Approach: Introducing MARIO, a novel method that addresses the key challenges in unsupervised OOD generalization, is a significant contribution.

(3) Extensive Experiments: The paper's comprehensive experimental analysis lends credibility to its findings and the effectiveness of MARIO in various scenarios.

Cons

(1) Clarity and Readability: The paper requires improved clarity, especially in section 3. The connection of Definition 2 and Theorem 3.1 to the context is unclear, necessitating better explanations and possibly pseudocode for illustrating the framework. Some used concepts and concrete equations are not well explained.

(2) Lack of Experimental Details: There is an absence of detailed information about the parameters used in baseline methods, which is crucial for replicability and understanding the experiments.

**Questions:**

(1) can the proposed objective be regarded as a plugin and also help to improve the supervised methods? If not, can more explanation be provided?

**Reviewer Confidence:**

2: The reviewer is willing to defend the evaluation, but it is likely that the reviewer did not understand parts of the paper

**Scope:**

4: The work is relevant to the Web and to the track, and is of broad interest to the community

---

### Decision · Program_Chairs · 2024-01-22

**Decision:**

Accept

**Comment:**

**Meta-review**: This paper proposes MARIO, a recipe for improving the adaptability of graph contrastive learning methods. Reviewers generally liked the paper, and the discussion was generally productive with several reviewers revising their ratings.

 **Strengths**:
 + Relevance of study (S5XT)
 + Interesting theoretical connection (oqvZ)
 + Extensive experimentation (oqvZ,Bco1)

 **Weaknesses**: *mostly addressed during discussion*
 - Writing needs improvement (S5XT, Bco1)